

# Satellite quantification of methane emissions and oil/gas methane intensities from individual countries in the Middle East and North Africa: implications for climate action

Zichong Chen[1], Daniel J. Jacob[1], Ritesh Gautam[2], Mark Omara[3], Robert N. Stavins[4], Robert C. Stowe[4], Hannah O. Nesser[1], Melissa P. Sulprizio[1], Alba Lorente[5], Daniel J. Varon[1], Xiao Lu[6], Lu Shen[7], Zhen Qu[1], Drew C. Pendergrass[1], and Sarah Hancock[1]

[1]School of Engineering and Applied Sciences, Harvard University, Cambridge, MA, USA

[2]Environmental Defense Fund, Washington DC, USA

[3]Environmental Defense Fund, Austin, TX, USA

[4]John F. Kennedy School of Government, Harvard University, Cambridge, MA, USA

[5]SRON Netherlands Institute for Space Research, Leiden, the Netherlands

[6]School of Atmospheric Sciences, Sun Yat-sen University, Zhuhai, Guangdong, China

[7]Department of Atmospheric and Oceanic Sciences, School of Physics, Peking University, Beijing, China

*Correspondence to*: Zichong Chen (zchen1@g.harvard.edu)

**Abstract.** We use 2019 TROPOMI satellite observations of atmospheric methane in an analytical inversion to quantify methane emissions from the Middle East and North Africa at up to ~25 km × 25 km resolution, using spatially allocated national UNFCCC reports as prior estimates for the fuel sector. Our resulting best estimate of anthropogenic emissions for the region is 35% higher than the prior bottom-up estimate (+103% for gas, +53% for waste, +49% for livestock, -14% for oil) with large variability across countries. Oil and gas account for 38% of total anthropogenic emissions in the region. TROPOMI observations can effectively optimize and separate national emissions by sector for most of the 23 countries in the region, with 6 countries accounting for most of total anthropogenic emissions including Iran (5.3 (5.0-5.5) Tg a$^{-1}$; best estimate and uncertainty range), Turkmenistan (4.4 (2.8-5.1) Tg a$^{-1}$), Saudi Arabia (4.3 (2.4-6.0) Tg a$^{-1}$), Algeria (3.5 (2.4-4.4) Tg a$^{-1}$), Egypt (3.4 (2.5-4.0) Tg a$^{-1}$) , and Turkey (3.0 (2.0-4.1) Tg a$^{-1}$). Most oil/gas emissions are from the production (upstream) subsector, but Iran, Turkmenistan, and Saudi Arabia have large gas emissions from transmission and distribution subsectors. We identify a high number of annual oil/gas emission hotspots in Turkmenistan, Algeria, Oman, and offshore in the Persian Gulf. We show that oil/gas methane emissions for individual countries are not related to production, invalidating a basic premise in the construction of activity-based bottom-up inventories. Instead, local infrastructure and management practices appear to be key drivers of oil/gas emissions, emphasizing the need for including top-down information from atmospheric observations in the construction of oil/gas emission inventories. We examined the methane intensity, defined as the upstream oil/gas emission per unit of methane gas produced, as a measure of the potential for decreasing emissions from the oil/gas sector, and using as reference the 0.2% target set by industry. We find that the methane intensity in most countries is considerably higher than this target, reflecting leaky infrastructure combined with deliberate venting or incomplete flaring of gas. However, we also find that Kuwait, Saudi Arabia, and Qatar meet the industry target and thus show that the target is achievable through capture of associated gas, modern infrastructure, and concentration of operations. Decreasing





methane intensities across the Middle East and North Africa to 0.2% would achieve a 90%
decrease in oil/gas upstream emissions and a 26% decrease of total anthropogenic methane
emissions in the region, making a significant contribution toward the Global Methane Pledge.

1. Introduction

Methane ($CH_4$) is a potent greenhouse gas with a relatively short atmospheric lifetime of 9.1 ±
0.9 years (Prather et al., 2012; Naik et al., 2021) and is a precursor of tropospheric ozone (Fiore
et al., 2002). Decreasing methane emissions is a powerful lever to mitigate near-term warming
(Naik et al., 2021), and thereby give the world time to "bend the curve" on carbon dioxide ($CO_2$)
emissions and removal, as well as to adapt to climate change. Anthropogenic emissions of
methane are from many sectors including the oil and gas supply chain, coal mining, livestock,
rice cultivation, landfills, and wastewater treatment. Natural emissions are mainly from wetlands.
Improving knowledge of methane emissions is urgently needed for enforcing the enhanced
transparency framework of the Paris Agreement, and the 2023 objectives of the Global Methane
Pledge (Climate and Clean air Coalition, 2021). The 194 Parties to the Paris Agreement
(individual nations plus the European Union) of the United Nations Framework Convention on
Climate Change (UNFCCC) have each submitted their periodic Nationally Determined
Contributions (NDCs), indicating how much they expect to reduce their greenhouse gas
emissions by specific years, most often by 2030. Emission inventories reported by Parties under
the Paris Agreement typically rely on bottom-up estimates using activity data and emission
factors that are extrapolated from limited information and may have large errors (Kirschke et al.,
2013; Saunois et al., 2020; Nisbet et al., 2020). Top-down methods involving inversion of
atmospheric methane observations can reduce these uncertainties through Bayesian synthesis
(Houweling et al., 2017). Here we use an inverse analysis of 2019 satellite observations of
atmospheric methane to quantify emissions by sector over the Middle East and North Africa
region including 23 individual countries.

The Middle East and North Africa is a compelling target region for reducing methane emissions
because of intense oil and gas production activity, contributing 32% to global oil production and
24% to global gas production in 2019 (EIA, 2020). The oil/gas sector presents the largest low-
cost mitigation potential for methane emissions with technically feasible solutions (Nisbet et al.,
2020). National inventories reported to the UNFCCC give a total oil/gas methane emission from
the Middle East and North Africa of 13.0 Tg a$^{-1}$ for 2019, representing 27% of global emissions
from that sector (Scarpelli et al., 2022). However, emission uncertainties are particularly high for
the oil/gas sector because of the large number of point sources with widely variable operating
conditions. Bottom-up estimates for individual countries may vary by more than an order of
magnitude (Scarpelli et al., 2022). Satellite observations have detected exceedingly large point
sources from oil and gas fields in the Middle East and North Africa (Varon et al., 2019, 2021;
Guanter et al., 2021; Lauvaux et al., 2022; Irakulis-Loitxate et al., 2022a, 2022b; Sánchez-García
et al., 2022; Ehret et al., 2022), revealing poor maintenance practices and equipment failures that
would likely not be accounted for in the bottom-up inventories.

Top-down emission estimates using atmospheric methane observations offer an independent
check on bottom-up inventories. They generally involve inverse analysis in which an
atmospheric transport model is used to relate emissions to atmospheric concentrations, equipped



with prior information from a spatially resolved emission inventory. Comparing the predicted atmospheric concentrations from the prior emission inventory to the observations enables correction of the inventory by Bayesian synthesis (Brasseur and Jacob, 2017). Satellite observations in the shortwave infrared (SWIR) are particularly attractive for top-down analyses
due to their global coverage and sensitivity down to the surface (Jacob et al., 2016, 2022). Inversions with satellite observations from the Greenhouse Gases Observing Satellite (GOSAT) for 2009-present (Qu et al., 2021; Deng et al., 2022) have enabled an assessment of national emissions across the globe in support of the Paris Agreement's Global Stocktake process (Worden et al., 2022), the first of which is to be completed in 2023. But the GOSAT observations are sparse, separated by about 250 km, which limits the spatial resolution that can be achieved and introduces errors in attributing emissions to countries and sectors. The TROPOspheric Monitoring Instrument (TROPOMI) satellite instrument (2018-present) provides global continuous daily mapping of atmospheric methane at 7 km × 5.5 km nadir resolution. It has unique capability for high-resolution quantification of national emissions and effectively
attributing emissions to sectors. This capability has recently been demonstrated for North America (Zhang et al., 2020; Shen et al., 2021, 2022) and East Asia (Chen et al., 2022; Liang et al., 2022).

Here we use TROPOMI observations for 2019 with the GEOS-Chem atmospheric transport model in an analytical inversion including closed-form error characterization to infer methane emissions from the Middle East and North Africa (-20º-70º E, 12º-44º N) at up to the native 0.25º×0.3125º (~ 25 × 25 km$^2$) resolution of GEOS-Chem. This allows us to quantify emissions by sector for 23 individual countries across the region and compare to the UNFCCC-reported inventories used as prior estimates in our inversion. We infer methane intensities (emissions per unit gas production) from the oil/gas sector in different countries and identify high-intensity
countries with the potential to greatly reduce emissions.

2. Data and Methods

2.1 TROPOMI satellite observations

TROPOMI is onboard the polar sun-synchronous Sentinel 5 Precursor satellite with a ~13:30 local overpass time. Dry column methane mixing ratios ($X_{CH4}$) are retrieved with a full-physics algorithm in the 2.3 μm absorption band with a global success rate of 3% over land limited by cloud cover and by dark or heterogeneous surfaces (Lorente et al., 2021). TROPOMI provides full global daily coverage with a spatial resolution of 7 km × 5.5 km in the nadir (7 km × 7 km before August 2019) (Hu et al., 2016). We use the TROPOMI methane product version 2.02 from the Netherlands Institute for Space Research (Lorente et al., 2021) for 2019 excluding low-
120 quality retrievals ('qa_value'<0.5) and snow-covered scenes identified with a blended albedo exceeding 0.8 (Chen et al., 2022).

The TROPOMI $X_{CH4}$ data can be affected by retrieval artifacts correlated with SWIR surface albedo also retrieved by TROPOMI (Barré et al., 2021). Here we apply a bias correction to TROPOMI retrievals over the Middle East and North Africa by calibrating to GOSAT observations. GOSAT has higher spectral resolution than TROPOMI and retrieves $X_{CH4}$ in the 1.65 μm band using the $CO_2$ proxy retrieval method, which is less subject to retrieval artifacts (Parker et al., 2019). We find that the differences between TROPOMI and GOSAT retrievals averaged on the 0.25º×0.3125º GEOS-Chem grid have a linear dependence on SWIR surface albedo (Fig. 1), and we apply the linear regression as a correction to the TROPOMI data. The



correction includes a non-zero intercept of 10.3 ppb but this is of no consequence because the same correction is applied to the initial and boundary conditions for the inversion. The mean TROPOMI-GOSAT difference on the 0.25°×0.3125° grid is -0.01 ± 9.3 ppb after this correction, where the standard deviation refers to the spatial variability of the annually averaged differences. This standard deviation, which is a measure of variable bias, is below the threshold requirement of 10 ppb by Buchwitz et al (2015) for satellite data to be effective in regional inversions.

Fig. 2 shows the spatial distribution of the corrected TROPOMI observations and the number of successful retrievals for 2019. The total number of TROPOMI retrievals over our inversion domain for 2019 is 30366339, evenly distributed across seasons. We average the TROPOMI retrievals (including $X_{CH4,}$ prior vertical profiles, and averaging kernel vectors) over each GEOS-

140 Chem 0.25°×0.3125° grid cell and each hour to yield $m$ = 3714062 super-observations for use in the inversion.

## 2.2 Prior emissions

Fig. 3 shows the distribution of prior emissions by sector over the inversion domain, Table 1 lists the domain-wide totals, and Table 2 lists totals for individual countries. Oil, gas, and coal emissions are from the Global Fuel Exploitation Inventory (GFEIv2), which uses detailed infrastructure data to spatially allocate on a 0.1°×0.1° grid the national inventories from individual countries reported to the UNFCCC including offshore emissions (Scarpelli et al., 2022). Iraq, Algeria, and Oman have not reported their emissions to the UNFCCC since 2000, and for those countries GFEIv2 uses recommended emission factors from the IPCC (2006) Tier 1

method and EIA production statistics (EIA, 2020) to infer national emissions. For other anthropogenic sectors (livestock, landfills, wastewater treatment, rice, and other minor sources), prior emissions are from the EDGARv6 inventory for 2018 (Crippa et al., 2021).

Wetland emissions are 2019 monthly means at 0.5°×0.5° resolution from the nine high-performance members of the WetCHARTS v1.3.1 inventory ensemble, so chosen because they fit best to a global GOSAT inversion (Ma et al., 2021). Other natural sources include open-fire emissions from the Global Fire Emissions Database version 4s (GFED4s) (van der Werf et al., 2017), termite emissions from Fung et al., (1991), and geological seepage emissions from Etiope et al. (2019) with global scaling to 2 Tg a$^{-1}$ (Hmiel et al., 2021). Termite emissions in the region are larger than wetlands (0.51 Tg a$^{-1}$ versus 0.42 Tg a$^{-1}$) and are mostly in Iran and Niger.

## 2.3 GEOS-Chem forward model

We use the nested version of the GEOS-Chem 13.0.0 chemical transport model (https://doi.org/10.5281/zenodo.4618180) as forward model for the inversion to relate methane emissions to atmospheric concentrations through atmospheric transport. GEOS-Chem is driven by meteorological fields from the GEOS-FP analyses (Lucchesi, 2018) at 0.25° × 0.3125° resolution. We use that native resolution in GEOS-Chem over the Middle East and North Africa domain (-20° -70° E, 12° - 44° N) with dynamic boundary conditions from a global model simulation using posterior methane emissions optimized from TROPOMI data following Shen et al (2022). We further optimize the boundary conditions for each quadrant (north, south, west, east) and for each season as part of the inversion. Initial conditions on 1 January 2019 are set to

170 match the mean TROPOMI column mixing ratios in the region following Qu et al (2021). In this manner, differences between the forward model and observations can be attributed to errors in 2019 emissions rather than to errors in initial conditions.





2.4 Analytical inversion procedure

We perform the inversion analysis mostly following Chen et al (2022). We use the Gaussian mixture model (GMM) of Turner and Jacob (2015) to define the state vector $\boldsymbol{x}$ of the inversion as emission patterns that TROPOMI observations can effectively constrain, aiming to preserve native (0.25°× 0.3125°) resolution for strong localized sources while smoothing the solution in regions with weak uniform emissions as provided in the prior knowledge. In the GMM, similarity vectors defining proximity and commonality in sectoral emissions (as defined by the prior estimate) are used to construct Gaussian state vector elements characterized by location of maximum emission, spatial standard deviation, and emission amplitude. Here we add as similarity vector the list of ultra-emitters (>25 tons h⁻¹) identified by Lauvaux et al (2022) from analysis of hotspots in the 2019-2020 TROPOMI data. This ensures that the ultra-emitters are resolved on the native 0.25°× 0.3125° grid of the inversion. We choose to use 600 Gaussian functions to optimize in the emission state vector, based on our previous experience with the information content of regional inversions. The inversion optimizes the amplitude of each Gaussian. We also optimize 16 boundary conditions (four boundaries × four seasons) for a total of $n = 616$ state vector elements.

We perform the inversion with lognormal error probability density functions (pdfs) for prior emissions (Maasakkers et al., 2019; Lu et al., 2022a). This prevents unphysical negative emissions (Miller et al., 2014) and better captures the heavy tail of the emission distribution (Yuan et al., 2015; Zavala-Araiza et al., 2015; Duren et al.,2019; Cusworth et al., 2022) than a normal error assumption. Specifically, we optimize $\ln(\boldsymbol{x})$ instead of $\boldsymbol{x}$, such that the prior errors on $\ln(\boldsymbol{x})$ (referred to hereafter as $\boldsymbol{x}'$) follow a normal distribution. The boundary condition elements of the state vector are still optimized assuming normal error distributions.

The inversion finds the optimal estimate of $\boldsymbol{x}'$ assuming normal error distributions (lognormal in emission space) by minimizing the Bayesian cost function $J(\boldsymbol{x}')$ (Brasseur and Jacob, 2017):

$$J(\boldsymbol{x}') = (\boldsymbol{x}' - \boldsymbol{x}'_a)^{\mathrm{T}}\mathbf{S}_a'^{-1}(\boldsymbol{x}' - \boldsymbol{x}'_a) + \gamma(\boldsymbol{y} - \mathbf{K}'\boldsymbol{x}')^{\mathrm{T}}\mathbf{S}_o^{-1}(\boldsymbol{y} - \mathbf{K}'\boldsymbol{x}') \tag{1}$$

where $\boldsymbol{x}' = \ln(\boldsymbol{x})$ and $\boldsymbol{x}'_a = \ln(\boldsymbol{x}_a)$, $\boldsymbol{x}_a$ $(n \times 1)$ is the prior emission estimate ($n = 616$), and $\boldsymbol{y}$ $(m \times 1)$ is the ensemble of TROPOMI super-observations ($m = 3714062$). $\mathbf{S}_a'$ $(n \times n)$ is the prior error covariance matrix and $\mathbf{S}_o$ $(m \times m)$ is the observational error covariance matrix, both assumed to be diagonal in absence of better objective information. $\mathbf{K}'\boldsymbol{x}' = \mathbf{K}\boldsymbol{x}$ is the GEOS-Chem forward model simulation of $X_{CH4}$. $\mathbf{K} = \partial\boldsymbol{y}/\partial\boldsymbol{x}$ $(m \times n)$ is the Jacobian matrix that describes the linear sensitivity of $\boldsymbol{y}$ to $\boldsymbol{x}$, and is constructed column by column by perturbing individual elements of $\boldsymbol{x}$ in GEOS-Chem. $\mathbf{K}' = \partial\boldsymbol{y}/\partial\boldsymbol{x}'$ $(m \times n)$ describes the sensitivity of $\boldsymbol{y}$ to $\boldsymbol{x}'$, which is nonlinear and readily derived from $\mathbf{K}$ following $\mathbf{K}'_{i,j} = \frac{\partial y_i}{\partial\ln(x_j)} = x_j\frac{\partial y_i}{\partial x_j} = x_j\mathbf{K}_{i,j}$, where $i$ and $j$ are indices of the observations and the state vector elements. The regularization factor $\gamma$ is introduced in Eq. (1) to prevent overfitting to observations because of the missing covariant structure (off-diagonal terms) in $\mathbf{S}_o$. We follow Lu et al (2021) and determine an optimal $\gamma$ value of 0.01 such that $\left(\widehat{\boldsymbol{x}}' - \boldsymbol{x}'_a\right)^{\mathrm{T}}\mathbf{S}_a'^{-1}\left(\widehat{\boldsymbol{x}}' - \boldsymbol{x}'_a\right) \approx n \pm \sqrt{2n}$, the expected value (±1 standard deviation) of the Chi-square distribution with $n$ degrees of freedom.

We solve the nonlinear optimization problem iteratively using the Levenberg-Marquardt method (Rodgers, 2000):



$$x'_{N+1} = x'_N + \left(\gamma {\mathbf{K}'_N}^{\mathbf{T}} \mathbf{S}_o^{-1} \mathbf{K}'_N + (1+\kappa)\mathbf{S}_a'^{-1}\right)^{-1} \left(\gamma {\mathbf{K}'_N}^{\mathbf{T}} \mathbf{S}_o^{-1}(y - \mathbf{K}x_N) - \mathbf{S}_a'^{-1}(x'_N - x'_a)\right) \quad (2)$$

where the coefficient $\kappa$ is fixed at 10 following Chen et al (2022), $N$ is the iteration number $(x'_0 = x'_a)$, and $\mathbf{K}'_N$ is evaluated for $x' = x'_N$. We iterate on Eq. (2) until the differences of all state vector elements between two consecutive iterations ($x'_N$ and $x'_{N+1}$) are smaller than 0.5%. We then take $\widehat{x}' = x'_{N+1}$ as the optimal posterior estimate.

The posterior error covariance matrix $\widehat{\mathbf{S}}'$ on the optimal posterior estimate is given by (Rodgers, 2000):

$$\widehat{\mathbf{S}}' = (\gamma \mathbf{K}'^{\mathbf{T}} \mathbf{S}_o^{-1} \mathbf{K}' + \mathbf{S}_a'^{-1})^{-1} \quad (3)$$

where $\mathbf{K}' = \mathbf{K}'_{N+1}$ is evaluated for the posterior estimate. The averaging kernel matrix $\mathbf{A}$ defining the sensitivity of the solution to the true value is given by

$$\mathbf{A} = \frac{\partial \widehat{x}'}{\partial x'} = \mathbf{I}_n - \widehat{\mathbf{S}}' \mathbf{S}_a'^{-1} \quad (4)$$

where $\mathbf{I}_n$ is the identity matrix. The trace of $\mathbf{A}$ quantifies the number of independent pieces of information on $x'$ obtained from the observations and is called the degrees of freedom for signal (DOFS).

An implication of using log-normal error statistics for emissions is that the inversion optimizes
the median (instead of the mean) of the lognormal emission pdf, but the mean can be inferred following $x_{mean} = x_{median} e^{\widehat{s}'/2}$, where $\widehat{s}'$ is the diagonal element of the posterior error covariance matrix (Eq. 3) corresponding to that emission sate vector element (Lu et al., 2022a). This is necessary when summing inversion results geographically such as to report national emissions.

2.5 Prior and observational error covariance matrices

We assume a geometric standard deviation factor ($\sigma_g = 2$) to characterize the lognormal error pdf for the prior emission estimates (i.e., the prior emissions are uncertain by a factor of 2) such that $\mathbf{S}_a'$ (with diagonal elements $s_a'$) is constructed following $\sqrt{s_a'} = \ln(\sigma_g)$ (Kirkwood, 1979). A factor of 2 is typical of the uncertainties in emission factors given by the IPCC for oil/gas
activities (Scarpelli et al., 2020). The prior error standard deviation on the boundary conditions is taken to be 10 ppb, which is typical of the root-mean-square error (RMSE) of GEOS-Chem simulations using posterior emission estimates (Chen et al., 2022).

We use the residual error method (Heald et al., 2004) to estimate observational error variances including contributions from the TROPOMI instrument, the retrieval, and the forward model. Here we take into account the error reduction resulting from averaging individual TROPOMI retrievals $y'$ into the super-observations $y$. We first apply the residual error method to individual retrievals in each 0.25°× 0.3125° grid cell $k$ over the course of 2019. The difference $y' - \mathbf{K}x_a$ between individual TROPOMI retrievals and the prior simulation is decomposed into an annual mean $\overline{y' - \mathbf{K}x_a}$ for that grid cell to be corrected in the inversion, and a residual ($y' - \mathbf{K}x_a -$
$\overline{(y' - \mathbf{K}x_a)}$) representing the observational error for $y'$. The variance $s_k$ of that observational error would populate the observational error covariance matrix if we ingested individual retrievals in the inversion, but in fact we ingest super-observations each representing an average





of $P$ individual retrievals. If the observational error for individual retrievals averaged into a super-observation was uncorrelated, then the observational error variance would decrease as $1/P$ (central limit theorem), but the decrease is less if the errors are correlated.

To estimate the observational error variance reduction associated with averaging $P$ retrievals into one super-observation for a given $0.25° \times 0.3125°$ grid cell and hour, we repeat the residual error method but now apply it to the super-observations $y$ instead of the individual retrievals. Instead of computing error variances for individual grid cells, we sort the errors by the number $P$ of individual retrievals that went into the super-observation and take statistics for the dependence of the observational error variance on $P$ over the whole inversion domain. Results are shown in Fig. 4 with comparison to the central limit theorem. We see that the decrease in the observational error variance with the number $P$ of individual retrievals going into a super-observation is much weaker than would be expected for uncorrelated errors. This implies that the observational errors for the individual retrievals contributing to a super-observation for a given $0.25° \times 0.3125°$ grid cell and hour are highly correlated. The forward model transport component of the observational error is in fact perfectly correlated because the model provides a single prediction for all individual retrievals. But most of the observational error is expected to be contributed by the satellite retrieval (Wecht et al., 2014), and it appears that this error component is also correlated between individual retrievals.

To model the observational error correlation between individual retrievals contributing to a super-observation and thereby fit the results of Fig, 4, we adopt a two-component error variance equation following Miyazaki et al (2012) and Pendergrass et al. (2022) to separate the contributions from the forward model transport error variance ($\sigma^2_{transport}$) and the satellite single-retrieval error variance ($\sigma^2_{retrieval}$) to the observational error variance of the super-observation ($\sigma^2_{super}$):

$$\sigma^2_{super} = \sigma^2_{retrieval}\left(\frac{1-r_{retrieval}}{P} + r_{retrieval}\right) + \sigma^2_{transport} \qquad (5)$$

Here $r_{retrieval}$ is the error correlation coefficient for the individual retrievals averaged into the super-observation, with the transport error being perfectly correlated ($r_{transport} = 1$) by definition. Fitting Eq. (5) to the data in Fig. 4 we obtain error standard deviations $\sigma_{retrieval} = 16.4$ ppb (with $r_{retrieval} = 0.55$) and $\sigma_{transport} = 4.5$ ppb. Some error correlation in retrievals would indeed be expected based on similarity in surface types and aerosol optical depth. The observational error standard deviation decreases initially as the number $P$ of averaged retrievals increases, and approaches an asymptotic value of 13.0 ppb for $P > 10$ including contributions from the transport error standard deviation and the super-observation retrieval error standard deviation ($r^{1/2}_{retrieval}\sigma_{retrieval} = 12.2$ ppb) added in quadrature. Validation of TROPOMI retrievals with ground-based column observations from the TCCON network by Lorente et al (2021) yields a retrieval error standard deviation of 13.3 ppb (11.5 ppb if excluding two high-latitude TCCON stations) when averaging all concurrent retrievals within 300 km of a TCCON station corresponding to 90-400 individual retrievals. This is in close agreement with our asymptotic value of 12.2 ppb. Our derived transport error standard deviation of 4.5 ppb for $X_{CH4}$ is consistent with the transport error standard deviation of 36 ppb for surface concentrations derived by Lu et al. (2021) from the residual error method at surface sites, considering that the amplitude of variability for column concentrations is about 10 times lower than for surface concentrations (Cusworth et al., 2018).



Using Eq. (5) for the dependence of the observational error variance on $P$ with fitted parameters, we can now adjust the observational error variances $s_k$ derived previously for individual retrievals in 0.25°× 0.3125° grid cells $k$ to apply to the super-observations actually ingested in the inversion. We define for this purpose a normalized scaling factor $g(P) = \sigma^2_{super}(P)/ \sigma^2_{super}(1)$ .

Thus a super-observation for grid cell $k$ in a given hour that averages $P$ retrievals has an observational error variance $g(P)s_k$. We construct the diagonal observational error covariance matrix $\mathbf{S_o}$ in this manner. The resulting observational error variance averages $(10.4\ ppb)^2$ for the super-observations in the inversion domain. The error correlation between individual retrievals suggests that there should be in fact some error correlation between super-observations, even though these observations are for different grid cells and/or different hours. This would introduce off-diagonal structure in $\mathbf{S_o}$ but we do not have sufficient information to construct this off-diagonal structure objectively. The regularization factor $\gamma$ in Eq. (1) is intended to account for this correlation and correct for the assumption of diagonality in $\mathbf{S_o}$, as explained above.

### 2.6 Attributing posterior emissions to individual countries and sectors

The posterior GMM state vector ($n \times 1$) can be readily mapped on the $p$ native 0.25°×0.3125° grid cells of the inversion domain using the GMM-generated weighting of each Gaussian on that grid as represented by a matrix $\mathbf{W_1}$ ($p \times n$). The contributions from each of $q$ emission sectors (Table 1) to the emissions in individual grid cells are taken from the prior inventories to produce a matrix $\mathbf{W_2}$ ($pq \times n$). We can then apply a summation matrix $\mathbf{W_3}$ ($r \times pq$) to aggregate emissions over $r$ countries and/or sectors of interest. The resulting matrix $\mathbf{W} = \mathbf{W_3}\mathbf{W_2}$ ($r \times n$) thus represents the linear transformation from the posterior GMM state vector ($n \times 1$) to a reduced state vector ($r \times 1$) of sectoral emissions from individual countries. The reduced state vector ($x_{red}$), posterior error covariance ($\hat{\mathbf{S}}_{\mathbf{red}}$), and averaging kernel matrix ($\mathbf{A}_{\mathbf{red}}$) are computed as

$$\hat{x}_{red} = \mathbf{W}\hat{x} \tag{6}$$
$$\hat{\mathbf{S}}_{\mathbf{red}} = \mathbf{W}\hat{\mathbf{S}}\mathbf{W^T} \tag{7}$$

$$\mathbf{A}_{\mathbf{red}} = \mathbf{W}\mathbf{A}\mathbf{W}^* \tag{8}$$

where $\mathbf{W}^* = (\mathbf{W^T}\mathbf{W})^{-1}\mathbf{W^T}$ is generalized pseudo-inverse of $\mathbf{W}$ (Calisesi et al., 2005).

### 2.7 Inversion ensemble and uncertainty estimate

Our base inversion described above makes assumptions on the values of inversion parameters including a factor of 2 uncertainty on the prior emissions (geometric error standard deviation $\sigma_g$ = 2), an error standard deviation $\sigma_b = 10$ ppb for boundary conditions, and a regularization factor $\gamma = 0.01$. The posterior error matrix of Eq. (3) is a fair representation of the uncertainty on the analytical solution ($\hat{x}'$, $\hat{\mathbf{S}}'$) given this choice of inversion parameters, but it does not account for

uncertainties in the parameters. We therefore generate a 36-member ensemble of sensitivity inversions varying the parameters. The inversion ensemble includes (1) using $\sigma_g = 1.5$ or 2.5, (2) using $\sigma_b =$ 5 or 20 ppb, (3) using $\gamma = 0.005$ or 0.02, and (4) assuming normal prior error distributions for emissions with an error standard deviation of 50% following Lu et al (2021). Similar to Chen et al. (2022), we find that the uncertainty range defined by the optimal estimates of this 36-member ensemble is larger than the posterior error from the base inversion. We thus report the uncertainty in posterior estimates as the range of solutions given by the inversion ensemble.



## 3. Results and Discussion

### 3.1 Evaluation of posterior emission estimates

Fig. 2 shows the posterior emissions and Fig. 5 shows the posterior/prior emission ratios on the 0.25º ×0.3125º grid. Also shown are the averaging kernel sensitivities (diagonal elements of the averaging kernel matrix **A**) that identify where the TROPOMI observations are most effective at quantifying emissions. We achieve 123 independent pieces of information (DOFS) to quantify emissions over the inversion domain. The GMM aggregates weak prior emissions mainly following spatial proximity (dictated by the similarity factors longitude and latitude). The rectilinear latitude-longitude patterns in low-emitting regions reflect this aggregation. Thin lines between some of the rectilinear patterns reflect the superimposition of corrections from individual Gaussians onto the 0.25º ×0.3125º grid.

We implemented the posterior emissions in GEOS-chem to check that the posterior simulation provides an improved fit to the TROPOMI observations as compared to the prior simulation (Fig. 5). The mean model bias over the inversion domain decreases from -10.4 to -0.31 ppb. The root-mean-square error (RMSE) decreases from 18.6 to 14.7 ppb, with improvement limited by the observational error (Fig. 4).

We also find an improved ability of the posterior estimate to fit to independent *in situ* surface flask measurements (Fig. 6). These *in situ* observations are collected from the GLOBALVIEWplus $CH_4$ ObsPack v4.0 database compiled by the National Oceanic and Atmospheric Administration (NOAA) Global Monitoring Laboratory (Schuldt et al., 2021). There are five sites in the region, most of them remote (Table S1). The overall mean bias across the five sites is reduced from -8.9 to -1.9 ppb. The RMSE decreases only slightly from 27.0 to 360 24.5 ppb, limited by the forward model transport error in simulating surface concentrations (Lu et al., 2021).

### 3.2 Emissions from individual countries and sectors

Table 1 gives the region-wide emissions over the Middle East and North Africa for 2019 including a total of 23 individual countries. Our best estimate of the posterior anthropogenic and natural emissions over this region are 38.6 and 1.6 Tg a$^{-1}$, respectively, as compared to 28.5 and 1.0 Tg a$^{-1}$ in the prior estimate. Oil/gas is the largest source (8.5 Tg a$^{-1}$ for oil and 6.3 Tg a$^{-1}$ for gas), followed by waste (13.2 Tg a$^{-1}$) and livestock (8.2 Tg a$^{-1}$). Waste includes emissions from landfills and wastewater, which are combined in the inversion because of their spatial overlap. Coal and rice emissions are minimal. Our best estimate of the total anthropogenic emissions in 370 the region is 35% higher than the prior estimate, which can be mainly attributed to upward corrections for gas (+3.2 Tg a$^{-1}$, +103%), waste (+4.6 Tg a$^{-1}$, +53%), and livestock (+2.7 Tg a$^{-1}$, +49%). We find a downward correction for oil (-1.4 Tg a$^{-1}$, -14%).

Table 2 gives the total and sectoral anthropogenic emissions for each of the 23 countries in the region. Also shown are averaging kernel sensitivities, which measure to what degree TROPOMI observations can quantify national emissions independently of the prior estimate (0 = not at all; 1 = fully). All countries have averaging kernel sensitivities greater than 0.65 except four with very low emissions. Our ability to separate emissions from individual countries in the inversion is shown in Fig.7 using error correlations between posterior national emission estimates (0 = perfect separation; ±1 = no separation). We find that most of the error correlations are smaller



than 0.2, indicating successful separation. Exceptions are between Palestine, Jordan, and Israel; between Syria and Iraq; and between the United Arab Emirates (UAE) and Oman, where our inversion shows limited confidence in separating the emission estimates by country (flagged in Table 2).

Our region-wide estimate of oil emissions is lower than in the UNFCCC-based estimate from GFEIv2 but individual countries may be either higher or lower. We find large downward corrections in Iran (-1.8 Tg a$^{-1}$), Iraq (-1.4 Tg a$^{-1}$), and Libya (-0.4 Tg a$^{-1}$), which we will see later are likely due to overestimate of emission factors used in the UNFCCC reports or from the IPCC (2006) Tier 1 method (Sect. 3.3). We find upward corrections in other major oil-producing countries, mainly in Oman (+1.4 Tg a$^{-1}$) and Turkmenistan (+1.0 Tg a$^{-1}$), likely due to the large

number of super-emitting point sources not accounted for in the UNFCCC estimates (Varon et al., 2019, 2021; Guanter et al., 2021; Lauvaux et al., 2022a, b; Ikakulis-Loixalte et al., 2022; Ehret et al., 2022).

We find upward corrections of gas emissions in all countries compared to the UNFCCC-based national bottom-up inventories as given by GFEIv2, mainly in Algeria (+1.0 Tg a$^{-1}$), Turkmenistan (+0.8 Tg a$^{-1}$), Saudi Arabia (+0.7 Tg a$^{-1}$), and Iran (+0.5 Tg a$^{-1}$). Again, this is likely due to super-emitting point sources not included in the reports. We further analyze gas emission by subsector (upstream, midstream, downstream) using gridded information from GFEIv2, and Table 3 shows results for the top emitting countries. The dominant subsector in Algeria is upstream (76%), while the dominant subsector in Iran is downstream (67%),

consistent with all of Iran's gas production being consumed domestically (EIA-Iran, 2021). Turkmenistan and Saudi Arabia also show high shares of downstream emissions (42 and 44%, respectively), reflecting their heavy domestic consumption. Saudi Arabia relies largely on its offshore production for domestic gas use (EIA, 2020), and transmission from offshore platforms to population centers likely explains the large contribution from midstream emissions (53%). The large difference in sub-sectoral contributions between countries stresses the importance of setting country-specific emission control strategies.

Figure 8 shows the hotspot 0.25°×0.3125° grid cells in our posterior estimate, defined by emissions greater than 2.0 tons h$^{-1}$ averaged over the year (18 Gg a$^{-1}$). Turkmenistan, Algeria, and Oman have a large number of hotspot grid cells, and for these countries we also estimate

exceedingly high national emissions from oil/gas activity (Table 2). The hotspots identified in our inversion have the same general geographical distribution as the ultra-emitting facilities (>25 tons h$^{-1}$) previously identified by Lauvaux et al (2022) from single-pass TROPOMI observations, as shown in Fig. 8, though the precise locations are often at odds. The Lauvaux et al. (2022) threshold for ultra-emitters is much higher than our threshold for hotspot grid cells because theirs is based on single-pass detection of emissions that may be single or intermittent events, whereas ours is based on annual mean emissions. This may also explain some of the differences in hotspot locations. We identify more hotspots in Saudi Arabia and Oman, where the single-point source detection method of Lauvaux et al (2022) is hindered by large regional enhancements. We also find a number of hotspots from offshore emissions in the Persian Gulf that they would not

have been able to detect with their method. Conversely, we detect no hotspots over Syria but Lauvaux et al (2022) detect several, likely reflecting poor prior information for Syria in our inversion.

3.3 Major emitting countries and comparison to previous studies



Fig. 9 compares our posterior emissions from the top six emitting countries in the Middle East and North Africa (Iran, Turkmenistan, Saudi Arabia, Algeria, Egypt, and Turkey, accounting for 62% of region-wide anthropogenic emissions) to the prior emission estimate and to previous inversion results from Worden et al (2022), Deng et al (2022), and Western et al (2021) that all used the much sparser GOSAT data. Worden et al. (2022) presented national results by mapping the global 2019 inversion results of Qu et al. (2021) at $2^o \times 2.5^o$ resolution, with prior estimate of fuel emissions from GFEIv1 (Scarpelli et al., 2020) and other sectors from EDGARv6. Deng et al (2022) collected a total of 11 independent inversions from different groups contributing to the Global Methane Budget initiative (Saunois et al., 2020) for 2010-2017. Western et al (2021) estimated total 2010-2017 emissions from North Africa at monthly $0.35^o \times 0.23^o$ resolution with prior emissions from GFEIv1 for fuel and the 2012 EDGARv4.3.2 inventory for other anthropogenic sources, but they did not separate their results by sectors.

Our estimate of total anthropogenic emissions in Iran is consistent with the prior estimate but with a shift in sectoral attribution from oil to livestock and waste (Table 2). Our oil/gas estimate (1.8 Tg a$^{-1}$) is within the large uncertainty range of Deng et al (2022) (1.0-6.2 Tg a$^{-1}$), but lower than Worden et al (2022) (3.1-4.3 Tg a$^{-1}$) and the UNFCCC-based GFEIv2 (3.1 Tg a$^{-1}$, our prior estimate). GFEIv2 uses emission factors obtained from the Iranian government report in 2000, likely unsuitable for 2019. The prior estimate of oil/gas emissions of Worden et al (2022) is from 2016 GFEI v1, higher than that in the updated 2019 GFEIv2, because of intensified economic sanctions beginning in 2018 (EIA-Iran, 2021). The $2^o \times 2.5^o$ resolution of Worden et al (2022) may also limit the inversion's ability to effectively separate emissions between Iran and Iraq, which are close to the border (Fig. 2).

Our estimate for Turkmenistan is higher than the prior emissions and on the high end of the uncertainty ranges from Worden et al (2022), and Deng et al (2022). We estimate higher oil/gas emissions (3.2 Tg a$^{-1}$) than GFEIv2 (1.4 Tg a$^{-1}$), pointing to dense super-emitting point sources that are not properly accounted for in the bottom-up estimates (Varon et al., 2019, 2021; Guanter et al., 2021; Lauvaux et al., 2022; Irakulis-Loitxate et al., 2022a; Ehret et al., 2022). Our oil/gas estimate is also at the high end of 0.9-2.8 Tg a$^{-1}$ of Deng et al (2022) and 2.0-3.2 Tg a$^{-1}$ of Worden et al (2022), which we explain by the better ability of TROPOMI than sparse GOSAT to capture point sources (Fig. 8).

Our estimate for Saudi Arabia is at the high end of the large Deng et al (2022) uncertainty range and is higher than the prior estimate and Worden et al (2022). We find that most of the emissions in Saudi Arabia are from waste. Our higher estimate than Deng et al. (2022) and Worden et al. (2022) likely reflects the low observational density of GOSAT over Saudi Arabia, as evidenced in in Worden et al. (2022) by very low averaging kernel sensitivities.

Our estimate of gas emissions in Algeria is smaller than Worden et al (2022), but our larger estimate of livestock and waste offsets gas and yields good agreement on the total national emission. We find a low error correlation (<0.2) between posterior gas and waste emissions in Algeria, implying that TROPOMI can effectively separate these two sectors. Both studies show much higher gas emissions than the prior estimate, reflecting point sources that are not accounted for in the UNFCCC-based inventory. Varon et al. (2021) found from repeated point-source sampling with the Sentinel-2 satellite instrument over a 10-month period that a single super-emitting oil well in Algeria amounted to 6% of the UNFCCC-reported oil/gas emissions.



Our estimate and Worden et al (2022) are in close agreement on the total and sectoral emissions for Egypt, featuring in our work a large increase of waste emissions over the prior estimate (Table 2). Western et al (2021) only reported a total emission for Egypt but stressed the underestimate of agricultural emissions (livestock + rice) in the national government report, consistent with our finding (Table 2) and Worden et al (2022). Both our work and Worden et al. (2022) show higher national total emissions than Western et al (2021) over Egypt and Algeria, which may reflect smoothing errors in the inability of GOSAT data to effectively inform their high-resolution inversion.

We attribute posterior emissions over Turkey largely to the livestock and waste sectors with little contribution from oil/gas, in contrast to the other five countries. Our estimate of total and sectoral emissions is lower than our prior estimate and consistent with Worden et al (2022).

3.4 Oil/gas emission factors, activity metrics, and methane intensities

IPCC (2006) recommends the use of emission factors and activity data to construct bottom-up emission inventories. For the upstream, midstream, and downstream oil/gas subsectors it recommends that emission factors be defined per unit of oil/gas produced, transported/stored, and consumed, respectively. For oil the emission is almost exclusively from the upstream subsector, while for gas all three subsectors can contribute (Table 3). Fig. 10 shows national upstream emission factors from major producers in the Middle East and North Africa, comparing our inversion results to GFEIv2, and using EIA oil and gas production statistics as activity metric (EIA, 2020). Also shown are the ranges from the IPCC (2006) Tier 1 guidelines. Scarpelli et al. (2022) pointed out that emission factors computed in this manner for the national inventories reported to the UNFCCC span several orders of magnitude, and our inversion finds the same. The IPCC (2006) Tier 1 emission factors themselves span two orders of magnitude (Fig. 10). Such a range means that the emission factors cannot be reliable, and further implies that production is not the appropriate activity metric for estimating methane emissions.

Fig. 11 further illustrates the unsuitability of predicting methane emissions from oil and gas production rates by showing the ranked production rates from the top-producing countries along with the corresponding posterior methane emissions. There is no significant relationship between the two. The largest emitters are not the largest producers. Recent studies in the US suggested that the number of wells and the drilling of new wells may be a better predictor of methane emission than production rates (Allen et al., 2022; Lu et al., 2022b; Varon et al., 2022). Enverus (2021) provides well counts in the Middle East and North Africa though the data are incomplete, particularly for new wells which could be the largest emitters (Allen et al., 2022). As shown in Table 4, we find that national emissions correlate weakly with well counts ($r = 0.25$ for oil and $r =0.19$ for gas), and the correlation increases only slightly when combined with production rates ($r = 0.26$-$0.28$).

It appears that the ability to relate oil/gas methane emissions to simple activity metrics is compromised by the importance of infrastructure type and management practices in driving oil/gas emissions. For example, national oil/gas emissions are largest in Algeria and Turkmenistan, due to the exceedingly leaky infrastructure previously documented by observations of dense point sources from space (Guanter et al., 2021; Varon et al., 2021; Lauvaux et al., 2022; Ehret et al., 2022). These dense point sources imply poor regulations and insufficient infrastructure (Lwaszczuk et al., 2021; Lauvaux et al., 2022; Irakulis-Loitxate et al.,



2022a). Long-lasting venting and leaks detected in Turkmenistan may be related to old and inefficient equipment (Carbon Limits, 2013; Varon et al., 2019; Irakulis-Loitxate et al., 2022a). Lack of infrastructure in Algeria to transport and process gas (Ouki., 2019) from remote production fields challenges the country's gas takeaway capacity, as illustrated by the exceedingly high volume of flared gas (Fig. 12) derived from flare radiances detected by the Visible Infrared Imaging Radiometer Suite (VIIRS) instrument (Elvidge et al., 2016). Studies in the US (Deighton et al., 2020; Omara et al., 2016, 2022) also found that equipment negligence and disrepair are the primary drivers of methane emissions for low production wells. The impact

of these stochastic processes (equipment maintenance, local management practices) on emissions is however difficult to quantify, and might vary largely from basin to basin, and from country to country. Construction of bottom-up inventories relies on activity metrics and is thus unable to accurately quantify oil/gas emissions. This finding hence stresses the critical importance of top-down emission estimates from atmospheric observations in the Global Stocktake and UNFCCC reporting in support of climate policy.

A useful metric for assessing the potential for emission reductions from the oil/gas industry is the methane intensity, defined by the industry-based Oil and Gas Climate Initiative (OGCI, 2021) as the upstream oil/gas emissions per unit of gas production. This measures the methane lost to the

530 atmosphere rather than taken to market. OGCI (2021) recently announced its 2025 upstream intensity target of 0.2%. Fig. 13 shows the methane intensities for major energy producing countries, assuming average methane gas content of 92% by volume (Scarpelli et al., 2022). We find a wide range of methane intensities across countries, spanning from 17.6% for Iraq to 0.06% for Qatar. The mean for the region is 1.8%, which can be compared to a mean value of 2.5% for the US in 2019 (Lu et al., 2022b). High methane intensities reflect leaky infrastructure combined with deliberate venting or flaring of gas. For example, emissions and production in 2019 are high in both Iraq and Iran, with the difference that gas is taken to market in Iran but vented/flared in Iraq, as indicated by the much higher ratio of VIIRS flared gas volume to gas production in Iraq (Fig. 12). This explains the higher methane intensity of Iraq compared to 0.61% for Iran. The

540 ratio of flared gas to production in Turkmenistan is even smaller than in Iran, but the methane intensity of Turkmenistan is much higher. This can be explained by prolonged venting and leaks (Varon et al., 2019; Irakulis-Loitxate et al., 2022a) related to poor infrastructure and management practices.

The OGCI (2021) methane intensity target of 0.2% is based on bottom-up emission models of methane emission from oil/gas infrastructure, and is vastly exceeded in all countries except three: Kuwait (0.15%), Saudi Arabia (0.14%), and Qatar (0.06%). There are likely one or more of the following reasons for their small intensities: (1) widespread associated gas capture. Saudi Arabia aims to capture most of its associated gas produced (EIA, 2020) and eliminate flaring by 2030 as

a part of the World Bank's Zero Routine Flaring Initiative; (2) modern infrastructure. More than half of Qatar's Liquefied Natural Gas (LNG) compressor mega trains, used to convert offshore gas to LNG, were built after 2009 (Qatargas, 2022). Also, Saudi Arabia has continuously invested in infrastructure to maintain its oil and gas production capacity (EIA, 2020); and (3) a small number of high-producing wells with centralized infrastructure. The majority of Qatar's gas is produced in ~200 wells in the offshore North Field, and processed in 14 compressor trains and two condensate refineries in Ras Laffan Industrial City (Qatargas, 2022). This finding suggests that infrastructure developments on improving associated gas capture, modernizing



equipment, and improving management practices are effective avenues to reducing the methane intensities and achieving the OGCI (2021) target of 0.2% methane intensity. Decreasing the methane intensities in all countries in the Middle East and North Africa to 0.2% would reduce total oil/gas upstream emissions in the region to 1.1 Tg a$^{-1}$ and represent a 26% reduction of total anthropogenic emissions in the region (Table 1). This would make a major contribution toward the collective goal of the Global Methane Pledge to decrease methane emissions by 30% by 2030 (Climate and Clean Air Coalition, 2021).

4. Conclusions

We used 2019 TROPOMI satellite observations in a high-resolution inversion to infer methane emissions from the Middle East and North Africa region at up to 25 ×25 km$^2$ resolution with emphasis on the contributions from individual countries and from the oil and gas sector. Our purpose was to evaluate the national inventories submitted to the United Nations Framework Convention on Climate Change (UNFCCC) under the Paris Agreement, and to identify avenues for climate action toward meeting the Global Methane Pledge.

Our inversion used as prior estimate a gridded version of the national fuel inventories reported by individual countries to the UNFCCC, thus enabling direct evaluation of these inventories. It applied Bayesian synthesis of the prior inventories with the TROPOMI observations to analytically obtain optimal emission estimates, thus providing closed-form characterization of information content and facilitating the creation of an inversion ensemble for conservative uncertainty estimates on posterior emissions. Innovations in our inversion methodology include specific resolution of ultra-emitters in the Gaussian mixture model (GMM) used as state vector for the inversion, and accounting for observational error correlation in the assimilation of TROPOMI observations.

We report optimized sector-resolved emissions for the 23 individual countries in the region. We find that TROPOMI observations can effectively constrain and individually separate emissions for most of the countries (19 out of 23). The others have small emissions. Total anthropogenic emissions in the region are 35% higher than in the prior estimate, reflecting increases in emissions from gas (+103%), waste (+53%), and livestock (+49%), but decrease for oil (-14%).

We find that the top six emitting countries, including Iran (5.3 (5.0-5.5) Tg a$^{-1}$, where numbers in parentheses are the range from our 36-member inversion ensemble), Turkmenistan (4.4 (2.8-5.1) Tg a$^{-1}$), Saudi Arabia (4.3 (2.4-6.0) Tg a$^{-1}$), Algeria (3.5 (2.4-4.4) Tg a$^{-1}$), Egypt (3.4 (2.5-4.0) Tg a$^{-1}$), and Turkey (3.0 (2.0-4.1) Tg a$^{-1}$) together make up 62% of the total anthropogenic emissions in the region. Oil and gas are major contributors to these emissions except for Turkey. Comparison of our results for these countries to previous inversions using GOSAT satellite data show some disagreements that may be related to the sparsity of GOSAT sampling. Most oil/gas emissions are from the upstream (production) subsector, but some countries including Turkmenistan, Saudi Arabia, and Iran have large gas emissions from midstream (transmission) and downstream (distribution) subsectors. We identify a number of emission hotspots (>18 Gg a$^{-1}$ on the 25×25 km$^2$ grid) particularly in Turkmenistan, Algeria, Oman, and offshore in the Persian Gulf. These hotspots are related to underestimates of oil/gas emissions in the national UNFCCC reports, indicating that they are not properly accounted for in the bottom-up inventories compiled for these reports.





The IPCC (2006) recommends the use of emission factors per unit oil or gas produced in the construction of bottom-up emission inventories, but these emission factors vary by orders of magnitude between countries and we find that there is in fact no significant relationship between emissions and production rates at the country level. Well counts are a better activity metric to predict emissions but national-scale correlations to emissions are still weak ($r = 0.19\text{-}25$), even in combination with production rates ($r = 0.26\text{-}0.28$). The importance and stochastic nature of local operating conditions and management practices in determining oil/gas emissions may stifle attempts to relate these emissions to simple activity metrics. This implies that top-down emission estimates from atmospheric observations are essential for the oil/gas sector and need to be considered as part of the Global Stocktake and UNFCCC reporting.

The potential to decrease methane emissions from the dominant upstream component of the oil/gas sector can be quantified by the methane intensity, defined by the Oil and Gas Climate Initiative (OGCI) industry consortium as the upstream oil/gas emissions per unit of gas production. The methane intensity measures the fraction of methane lost to the atmosphere rather than taken to market. OGCI has a target of reducing the methane intensity to 0.2% worldwide by 2025. We find that the methane intensities in almost all countries of the Middle East and North Africa are much larger, with highest values in Iraq (17.6 (7.5-30.6) %), followed by Oman (8.9 (4.5-13.4) %), Turkmenistan (4.6 (3.1-5.4) %), Libya (4.2 (2.6-8.2) %), UAE (3.3 (2.3-4.2) %), and Algeria (2.9 (1.8-4.4) %). These high values reflect leaky infrastructure combined with deliberate venting or flaring of gas. By contrast, we find that methane intensities in Kuwait (0.15%), Saudi Arabia (0.14%), and Qatar (0.06%) are lower than the OGCI target, demonstrating that this target is achievable. These three countries appear to achieve their low methane intensities through a combination of associated gas capture, modern infrastructure, and small number of high-producing wells with centralized processing. This suggests that modernization of infrastructure combined with associated gas capture and improved management practices can effectively reduce methane intensities elsewhere. Meeting the OGCI target of 0.2% methane intensity throughout the Middle East and North Africa would decrease oil/gas upstream emissions in the region by 90% and decrease total anthropogenic methane emissions in the region by 26%, making a significant contribution toward the Global Methane Pledge.



*Acknowledgments*. This work was funded by the Climate and Clean Air Coalition (CCAC) of the
United Nations Environment Programme (UNEP), by the Harvard University Climate Change
Solutions Fund (CCSF), and by the NASA Carbon Monitoring System.

*Data availability*. The TROPOMI satellite observations version 2.02 are available at
*http://www.tropomi.eu/data-products/methane*; The GOSAT methane retrievals version 9.0 are
available at *https://catalogue.ceda.ac.uk/uuid/18ef8247f52a4cb6a14013f8235cc1eb*;The
ObsPack GLOBALVIEWplus CH4 ObsPack v4.0 data product is available at
*https://gml.noaa.gov/ccgg/obspack/data.php*.

*Author contributions*. ZC and DJJ contributed to the study conceptualization. ZC conducted the
data and modeling analysis with contributions from RG, MO, RNS, RCS, HN, MPS, AL, DJV,
XL, LS, ZQ, DCP, and SH. ZC and DJJ wrote the paper with input from all authors.

*Competing interests*. The authors declare that they have no conflict of interest.



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



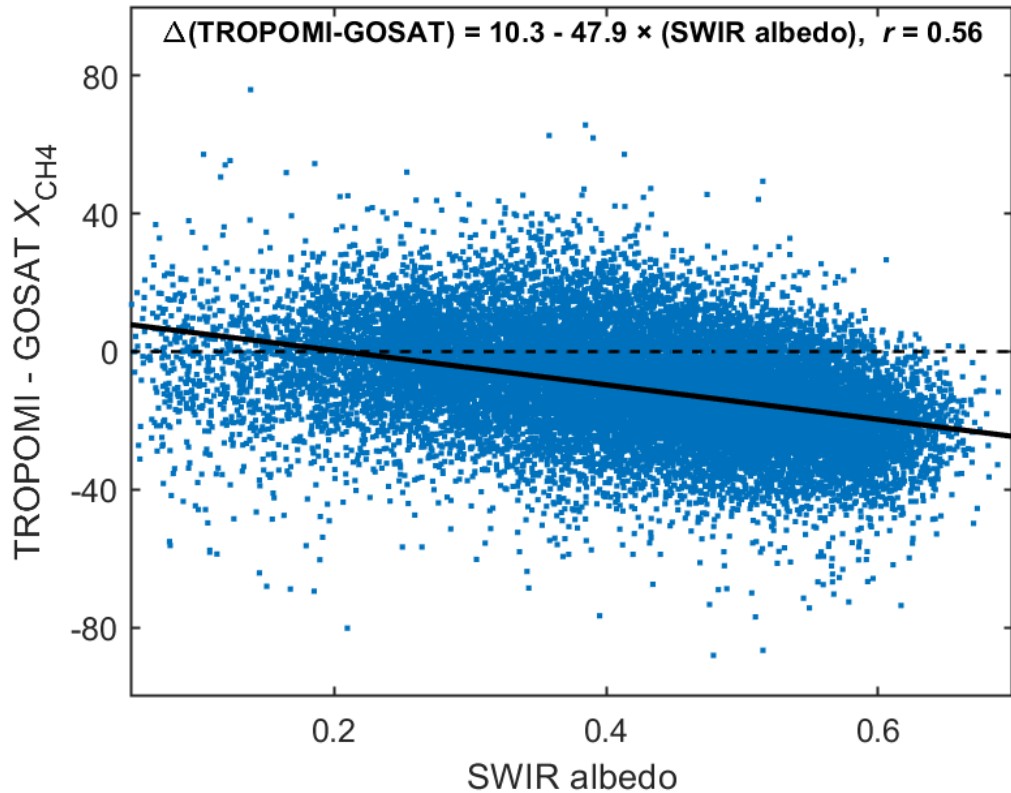

**Figure 1**. TROPOMI- GOSAT difference in retrieved dry column methane mixing ratio ($X_{CH4}$) as a function of the shortwave infrared (SWIR) surface albedo in the 2305-2385 nm range also retrieved by TROPOMI. Individual data points represent daily differences in collocated observations averaged on the GEOS-Chem 0.25°×0.3125° grid over the Middle East and North Africa (-20°-70° E, 12°-44° N) in 2019. The black solid line is the ordinary linear regression with coefficients given inset. The dashed line indicates zero difference between TROPOMI and GOSAT $X_{CH4}$.

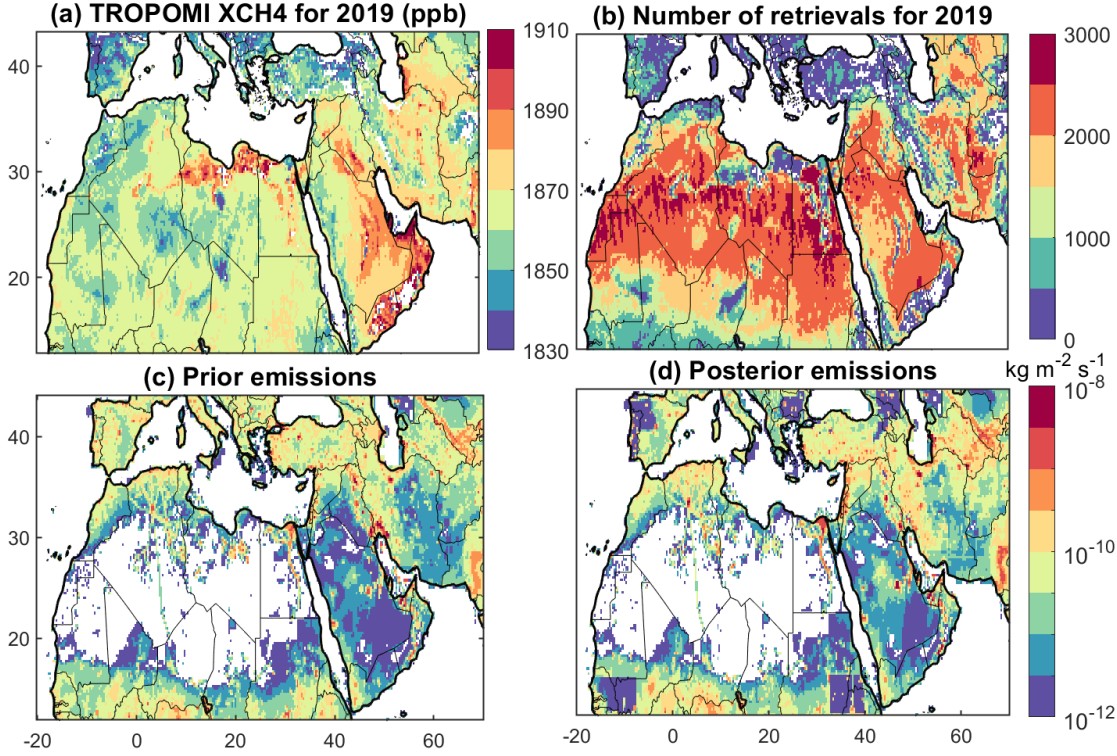

**Figure 2**. Atmospheric methane concentrations and emissions for the Middle East and North Africa. The top panels show the mean 2019 TROPOMI observations of dry column methane mixing ratio ($X_{CH4}$) and the total number of retrievals for that year on the 0.25°×0.3125° native grid of the inversion. The bottom panels show the prior and posterior emissions. Prior emissions are separated by sector in Fig. 3. Areas in blank have emissions lower than $1\times10^{-12}$ kg m$^{-2}$ s$^{-1}$.





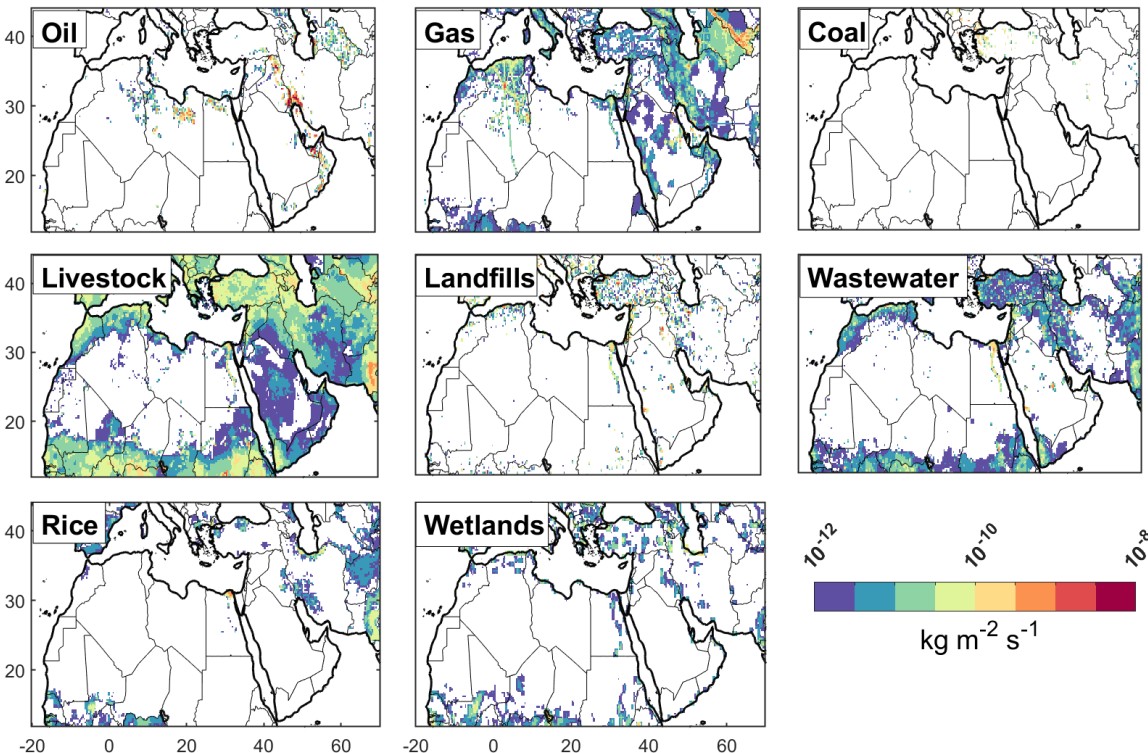

**Figure 3**. Prior estimates of methane emissions used for the inversion. Oil, gas, and coal emissions are from the GFEIv2 gridded version of the national inventories from individual countries reported to the UNFCCC (Scarpelli et al., 2022). Some countries do not report to the UNFCCC and their emissions are inferred by Scarpelli et al. (2022) from EIA production statistics. Other anthropogenic emissions are from EDGARv6 (Crippa et al., 2021). Wetland emissions are 2019 monthly means of the nine-member high-performance subset of the WetCHARTs inventory ensemble (Ma et al., 2021), and are shown here as the annual means. Emissions lower than $1\times10^{-12}$ kg m$^{-2}$ s$^{-1}$ are shown as blank. The total prior emission is shown in Fig. 2 and includes smaller sectors listed in Table 1.

1030





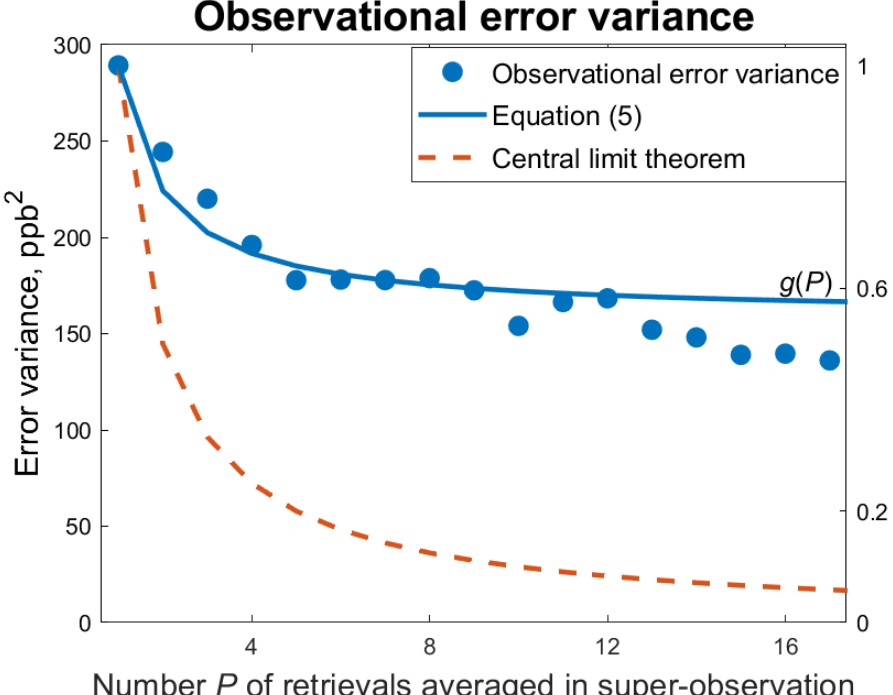

**Figure 4**. Reduction of the observational error variance ($\sigma^2_{super}$) from the averaging of individual TROPOMI retrievals into super-observations for ingestion in the inversion. The symbols show the error variances computed with the residual error method from individual super-observations over the Middle East and North Africa inversion domain as a function of the number $P$ of individual retrievals averaged into the super-observations. Each symbol represents the statistics for at least 100000 super-observations. The data are fitted to a two-component representation of the observational error variance $\sigma^2_{super}(P)$ as given by Eq. (5). Also shown in the Figure is the error variance reduction function $1/P$ if there was no error correlation between individual retrievals, as given by the central limit theorem.

1040



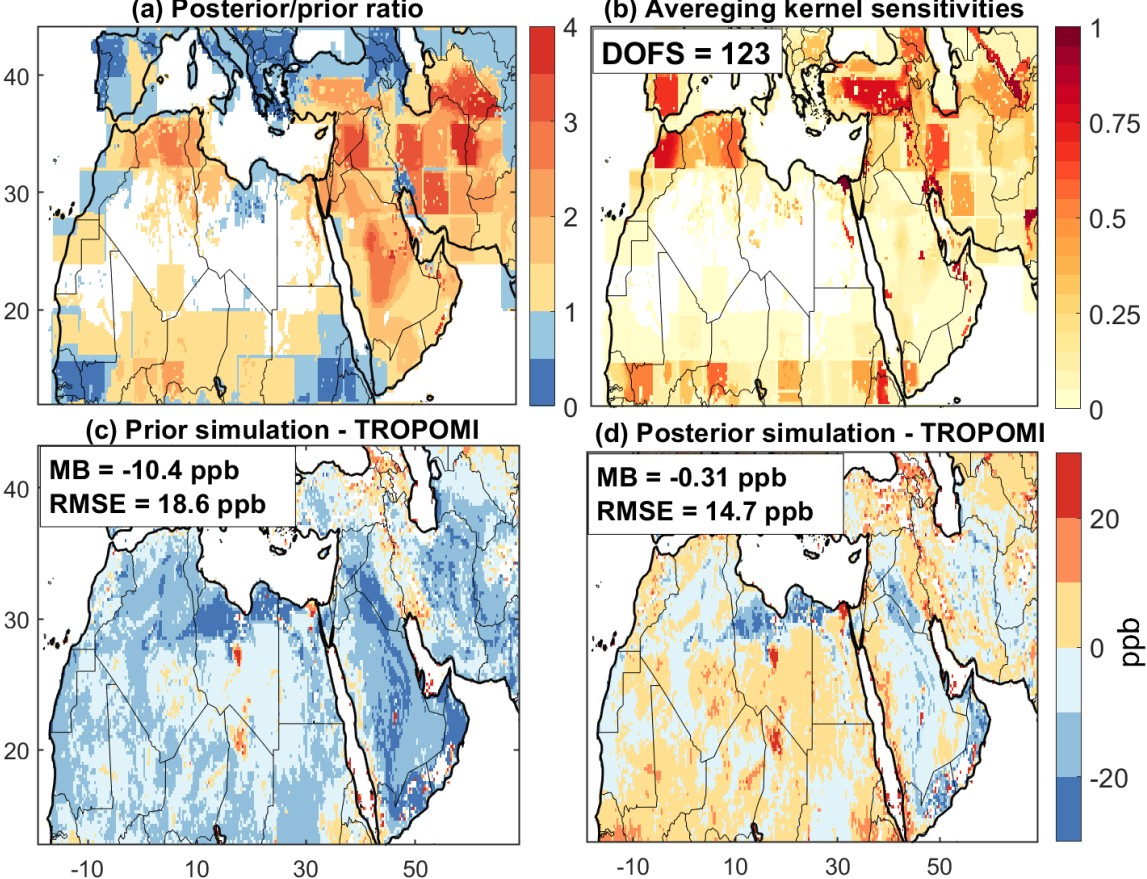

**Figure 5**. Optimization of methane emissions over the Middle East and North Africa in 2019 by inversion of TROPOMI observations. Results are from the base inversion and are shown on the 0.25°×0.3125° native grid of the inversion. (a) Ratios between posterior and prior emissions. (b) Averaging kernel sensitivities (dimensionless). The averaging kernel sensitivities are the diagonal elements of the averaging kernel matrix, indicating the ability of the observations to quantify emissions independently from the prior emissions (1 = fully, 0 = not at all). The number of degrees of freedom (DOFS, defined as the trace of the averaging kernel matrix) is given inset. (c) Mean differences between the GEOS-Chem simulation with prior emissions and observations. The mean bias (MB) and root-mean-square error (RMSE) over the study domain are given inset. (d) Same as (c) but for the GEOS-Chem simulation with posterior emissions.

1050



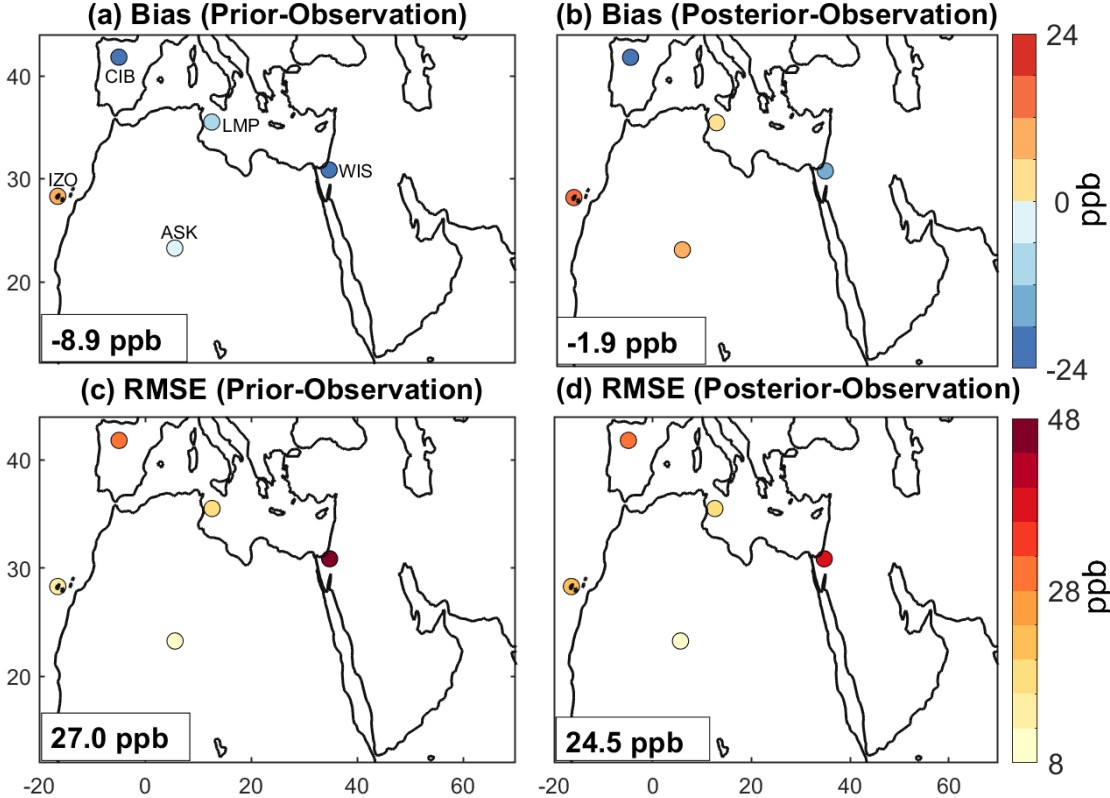

**Figure 6**. Evaluation of inversion results with independent *in situ* observations. The Figure compares GEOS-Chem simulations using prior or posterior emissions to *in situ* flask measurements from five surface sites in 2019 compiled by the NOAA GLOBALVIEWplus CH$_4$ ObsPack v4.0 database. The five sites are listed in Table S1. The annual mean biases and root-mean-square errors (RMSEs) for each site are shown. The spatial mean biases and the overall RMSEs for the ensemble of sites are given inset.



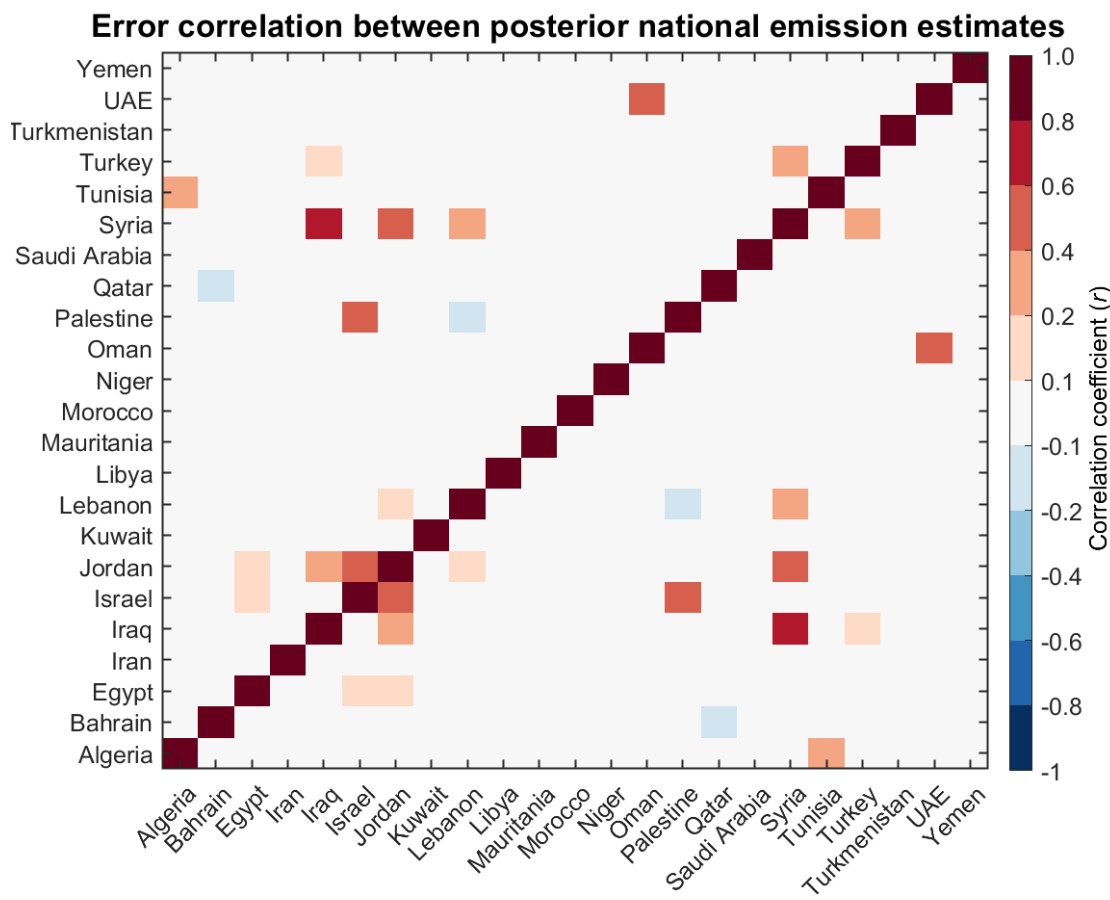

1070

**Figure 7**. Error correlation coefficients (*r*) between posterior estimates of total anthropogenic emissions from different countries in the inversion domain, measuring the ability of the inversion to separate emissions in one country from another (±1= not at all; 0 = fully).



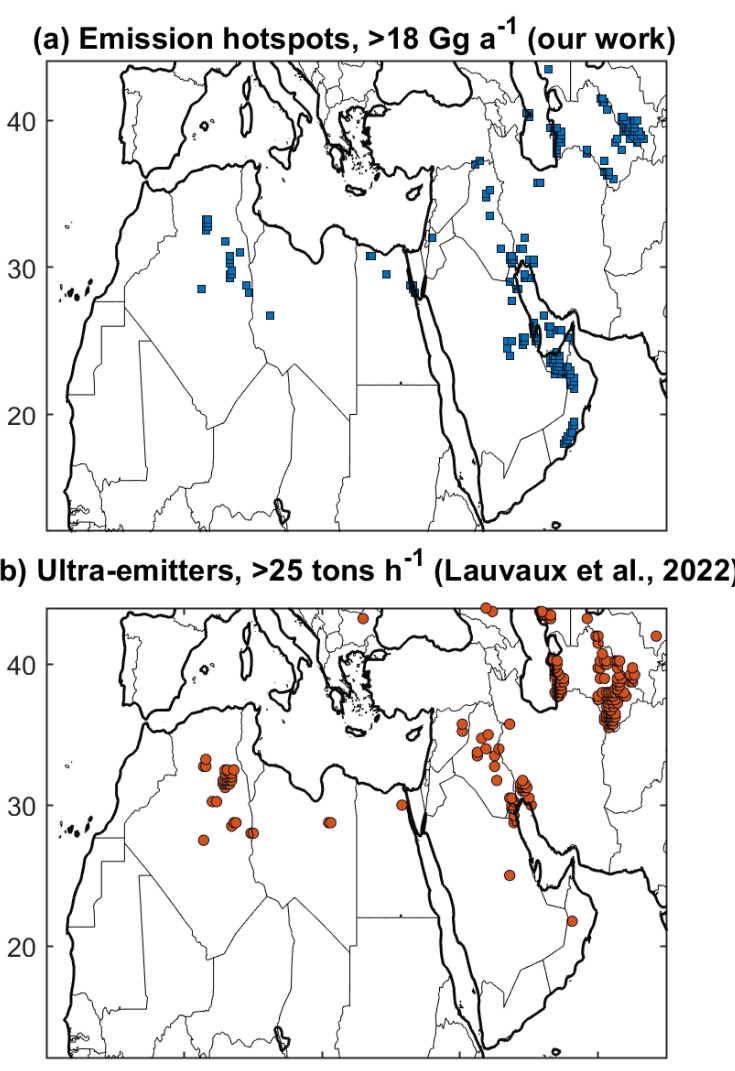

**Figure 8**. Methane emission hotspots from oil/gas activity in the Middle East and North Africa. Top panel shows the hotspot 0.25º × 0.3125º grid cells from our inversion in 2019, defined as emissions greater than 2.0 tons h$^{-1}$ averaged over the year (18 Gg a$^{-1}$). Bottom panel shows ultra-emitters (>25 tons h$^{-1}$) identified from 2019-2020 single-pass TROPOMI data by Lauvaux et al (2022).

1080



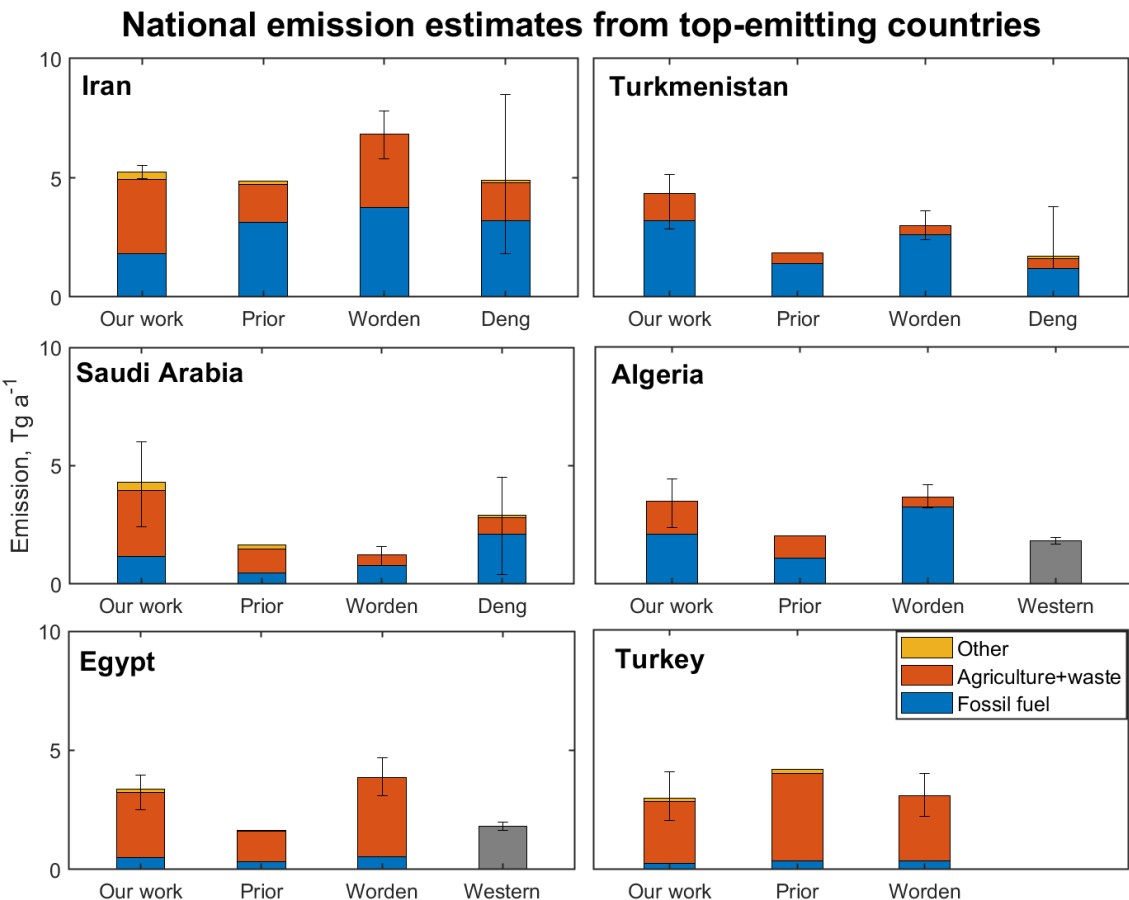

**Figure 9.** National methane emissions from the top six emitting countries in the Middle East and North Africa. Results from our work are compared to our prior estimates and to GOSAT inversions reported by Worden et al (2022), Deng et al (2022) (not for all countries), and Western et al (2021) (for North Africa only, without separation by sectors). Fossil fuel includes emissions from oil, gas, and coal; agriculture includes emissions from livestock and rice; and waste includes emissions from landfills and wastewater treatment. Vertical bars are reported uncertainty ranges for total national emissions. Sectors are aggregated to enable comparison with previous studies. More detailed sectoral breakdown for our work is in Table 2.





**Figure 10.** Country-level emission factors for oil and gas upstream activity in 2019. The emission factors represent the amount of methane emitted per unit of oil or gas produced, following the definition of IPCC (2006). Values are shown for our posterior estimates and for the UNFCCC reports as implemented in GFEIv2 and used as our prior estimate. Also shown is the range of values from the IPCC Tier 1 methods (IPCC, 2006), from the lowest value for developed countries to the highest value for developing countries and countries with economies in transition. GFEIv2 estimates of emission factors for Iraq, Oman, and Libya are from IPCC (2006) Tier 1 methods because these countries do not report to the UNFCCC. Horizontal bars indicate the uncertainty range inferred from our inversion ensemble.



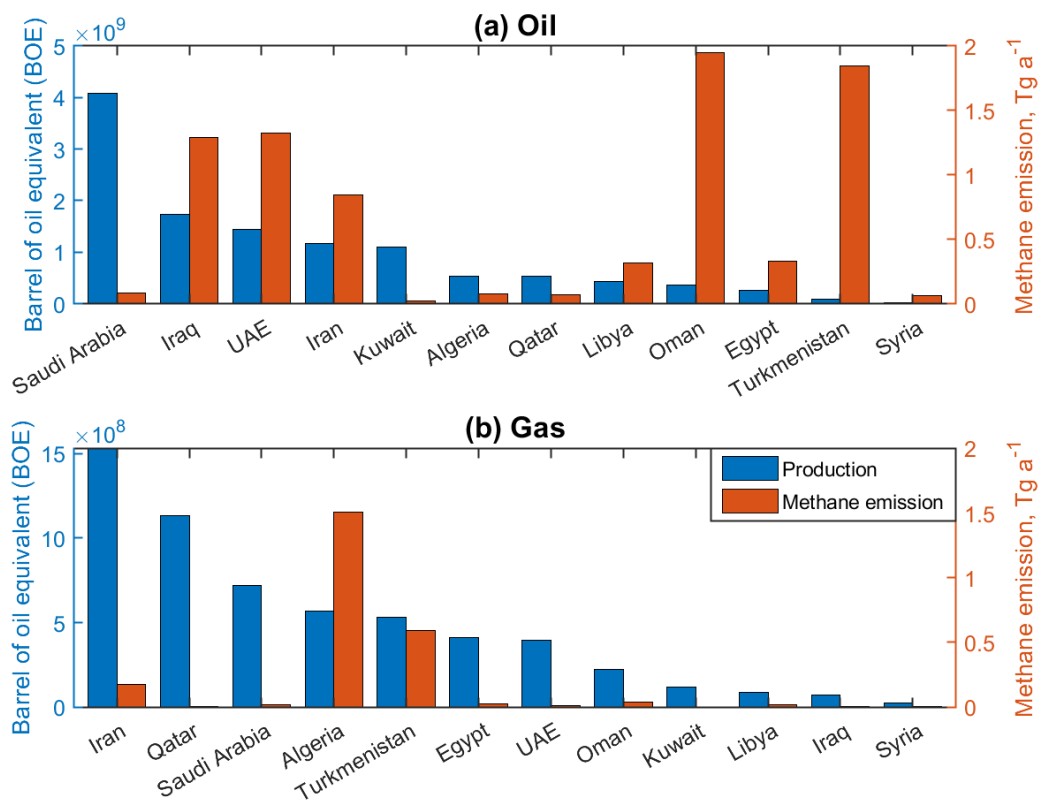

1100

**Figure 11**. Ranked oil and gas production rates in 2019 from the top-producing countries in the Middle East and North Africa, with corresponding posterior estimates of methane emissions from that sector. Production statistics are from EIA (2020).

1110



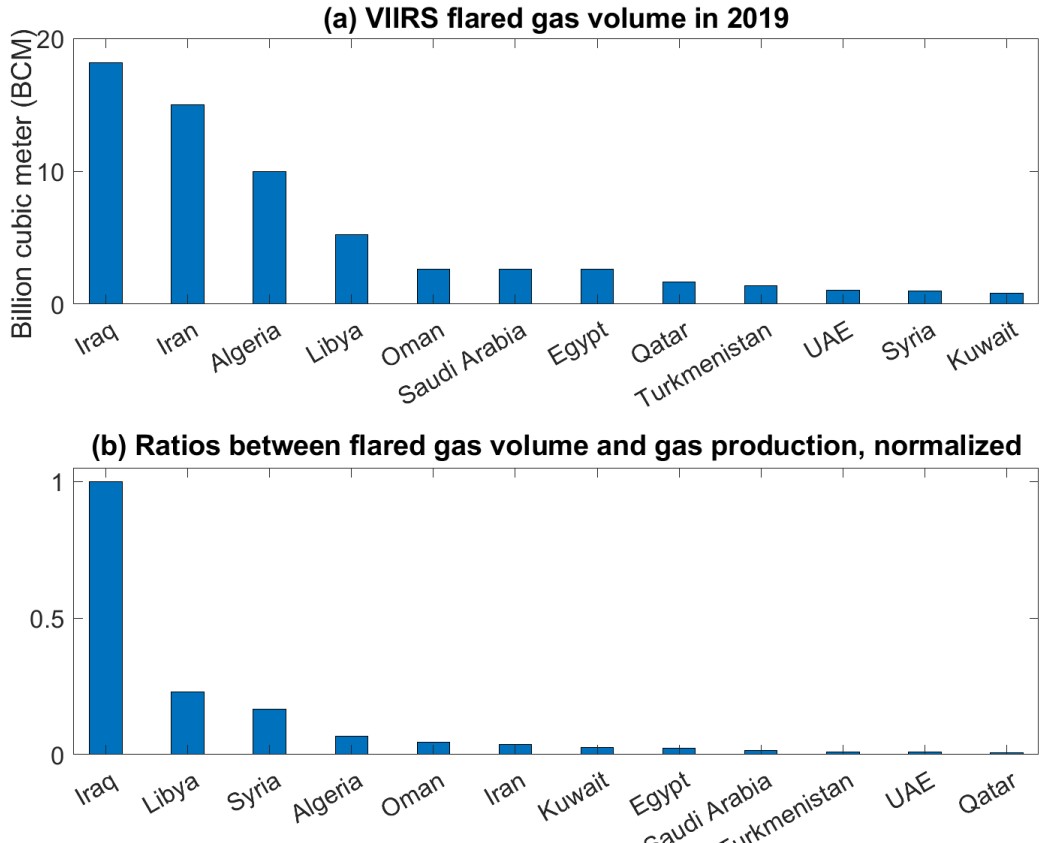

**Figure 12.** Flared gas volume in 2019 from major oil/gas producing countries. Top panel shows flared volume derived from flare radiances detected by the Visible Infrared Imaging Radiometer Suite (VIIRS) instrument. Bottom panel shows ratios of flared gas volume to gas production in 2019 normalized by a value of 264 m$^3$ flared gas per barrel of oil equivalent produced for Iraq.

1120



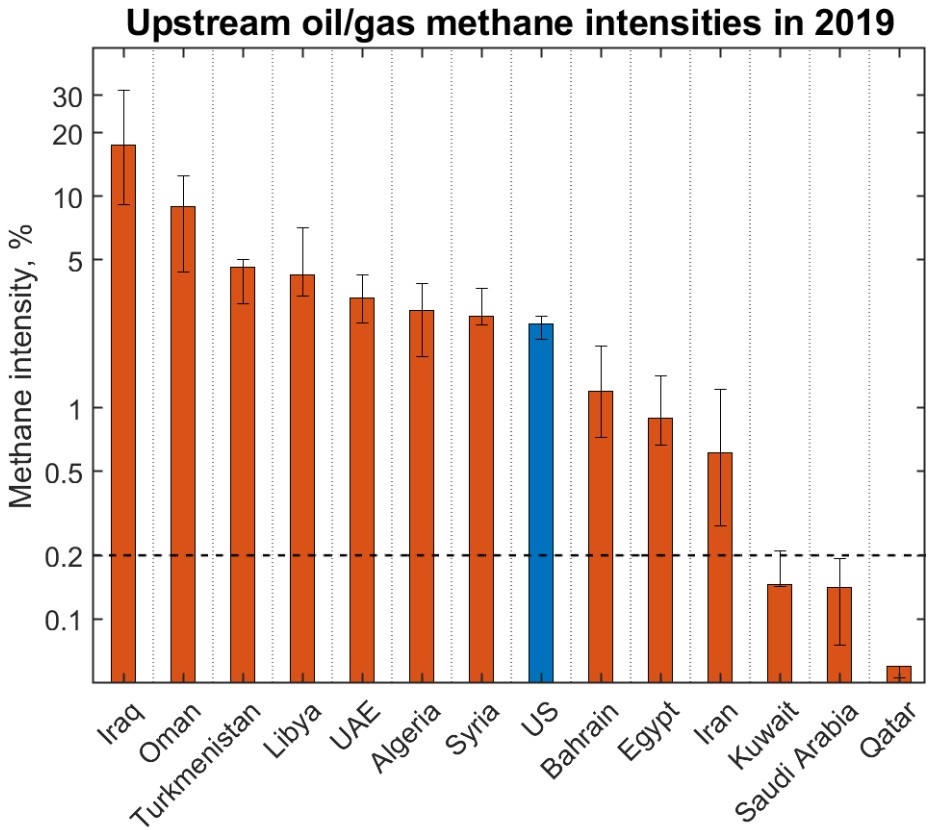

**Figure 13**. Methane intensities in 2019 for major oil/gas producing countries in the Middle East and North Africa. The methane intensity is defined as the amount of methane emitted from oil/gas upstream activities per unit of methane gas produced. Values are computed from our posterior emission estimates and EIA gas energy production statistics (EIA, 2020), assuming an average methane content of 92% by volume. Vertical bars indicate the uncertainty range inferred from our inversion ensemble. Dashed horizonal line indicates the OGCI (2021) industry target of 0.2% for 2025. Also shown is the mean and range of methane intensities from US oil/gas fields (Lu et al., 2022b).

1130





1140

**Table 1**. Methane emissions for 2019 in the Middle East and North Africa. [a]

|  | Prior estimate (Tg a$^{-1}$)[b] | Posterior estimate (Tg a$^{-1}$)[c] |
|---|---|---|
| **Total emission** | 29.5 | 39.7 (33.7-45.1) |
| **Anthropogenic** | 28.5 | 38.6 (32.5-43.2) |
| Oil | 9.9 | 8.5 (8.3-9.5) |
| Gas | 3.1 | 6.3 (4.1-8.3) |
| Livestock[d] | 5.5 | 8.2 (6.4-10.0) |
| Coal | 0.28 | 0.20 (0.10-0.27) |
| Waste[e] | 8.6 | 13.2 (10.6-14.2) |
| Rice | 0.38 | 0.72 (0.58-0.72) |
| Other[f] | 0.81 | 1.4 (1.0-1.6) |
| **Natural** | 1.0 | 1.6 (1.2-1.9) |
| Open fires | 0.02 | 0.03 (0.03-0.04) |
| Wetlands | 0.42 | 0.53 (0.49-0.54) |
| Seeps | 0.08 | 0.14 (0.10-0.18) |
| Termites | 0.51 | 0.87 (0.61-1.2) |

[a]Summing emissions over the 23 individual countries listed in Table 2.

[b] Prior estimates of oil, gas, and coal emissions are from the GFEIv2 gridded version of the national inventories from individual countries reported to the UNFCCC or inferred from EIA production data (Scarpelli et al., 2022). Other anthropogenic emissions are from EDGARv6. Wetland emissions are the mean of the high-performance subset of the WetCHARTs v1.3.1 inventory ensemble for 2019 (Ma et al, 2021). Open-fire emissions are from GFED4s (van der Werf et al., 2017). Termite emissions are from Fung et al (1991), and geological seepage emissions are from Etiope et al (2019) with scaling from Hmiel et al. (2020).

1150



[c]Results are from the base inversion of TROPOMI observations, with the uncertainty range in parentheses obtained from the 36-member inversion ensemble.

[d]Livestock sector includes contributions from enteric fermentation and manure management.

[e]Waste sector includes emissions from landfills and wastewater treatment, which are 5.2 and 3.4 Tg a[-1] in the prior estimate and are not separable in the inversion.

[f]Including industry, stationary combustion, mobile combustion, aircraft, composting, and field burning of agricultural residues.

1160

1170





**Table 2**. National anthropogenic methane emissions in 2019.[a]

| Country | | Oil | Gas | Livestock | Coal | Waste | Rice | Other | Anthropogenic total (Tg a$^{-1}$) | Sensitivity to observations[b] |
|---|---|---|---|---|---|---|---|---|---|---|
| **Algeria** | Posterior | 0.08 | 2.0 | 0.43 | 0 | 0.95 | 0 | 0.01 | 3.5 (2.4-4.4) | 0.84 |
| | Prior | 0.04 | 1.0 | 0.29 | 0 | 0.65 | 0 | 0.01 | 2.0 | |
| **Bahrain** | Posterior | 0.14 | 0.07 | 0 | 0 | 0.19 | 0 | 0.02 | 0.42 (0.39-0.43) | 0.91 |
| | Prior | 0.17 | 0.03 | 0 | 0 | 0.06 | 0 | 0.01 | 0.27 | |
| **Egypt** | Posterior | 0.35 | 0.15 | 0.78 | 0 | 1.6 | 0.35 | 0.12 | 3.4 (2.5-4.0) | 0.96 |
| | Prior | 0.24 | 0.07 | 0.41 | 0 | 0.71 | 0.15 | 0.06 | 1.7 | |
| **Iran** | Posterior | 0.78 | 1.0 | 1.20 | 0.04 | 1.7 | 0.22 | 0.32 | 5.3 (5.0-5.5) | 0.97 |
| | Prior | 2.6 | 0.52 | 0.65 | 0.02 | 0.81 | 0.13 | 0.14 | 4.9 | |
| **Iraq[c]** | Posterior | 1.2 | 0.04 | 0.17 | 0 | 0.73 | 0.02 | 0.05 | 2.2 (1.8-3.1) | 0.98 |
| | Prior | 2.9 | 0.03 | 0.14 | 0 | 0.54 | 0.01 | 0.04 | 3.7 | |
| **Israel[c]** | Posterior | 0 | 0.03 | 0.11 | 0 | 0.26 | 0 | 0.01 | 0.41 (0.31-0.41) | 0.81 |
| | Prior | 0 | 0.02 | 0.06 | 0 | 0.13 | 0 | 0 | 0.21 | |
| **Jordan[c]** | Posterior | 0 | 0.11 | 0.06 | 0 | 0.42 | 0 | 0.03 | 0.62 (0.45-0.68) | 0.91 |
| | Prior | 0 | 0.06 | 0.03 | 0 | 0.19 | 0 | 0.01 | 0.29 | |
| **Kuwait** | Posterior | 0.02 | 0.01 | 0.01 | 0 | 0.83 | 0 | 0.08 | 0.95 (0.58-0.98) | 0.85 |
| | Prior | 0.04 | 0 | 0.01 | 0 | 0.27 | 0 | 0.03 | 0.35 | |
| **Lebanon** | Posterior | 0 | 0.02 | 0.03 | 0 | 0.09 | 0 | 0 | 0.15 (0.09-0.18) | 0.76 |
| | Prior | 0 | 0.01 | 0.01 | 0 | 0.04 | 0 | 0 | 0.07 | |
| **Libya** | Posterior | 0.37 | 0.02 | 0.10 | 0 | 0.11 | 0 | 0.01 | 0.61 (0.56-0.85) | 0.76 |
| | Prior | 0.76 | 0.02 | 0.09 | 0 | 0.09 | 0 | 0.01 | 0.97 | |
| **Mauritania** | Posterior | 0 | 0.01 | 0.18 | 0 | 0.03 | 0 | 0.02 | 0.23 (0.20-0.28) | 0.20 |
| | Prior | 0 | 0.01 | 0.21 | 0 | 0.04 | 0.01 | 0.02 | 0.29 | |
| **Morocco** | Posterior | 0 | 0.04 | 0.61 | 0 | 0.67 | 0 | 0.02 | 1.3 (1.1-1.7) | 0.89 |
| | Prior | 0 | 0.03 | 0.38 | 0 | 0.71 | 0 | 0.02 | 1.2 | |
| **Niger** | Posterior | 0.01 | 0 | 1.1 | 0 | 0.18 | 0.01 | 0.08 | 1.3 (0.95-1.8) | 0.65 |
| | Prior | 0.01 | 0 | 0.69 | 0 | 0.12 | 0.01 | 0.05 | 0.86 | |
| **Oman[c]** | Posterior | 2.0 | 0.10 | 0.06 | 0 | 0.15 | 0 | 0.03 | 2.4 (1.2-3.4) | 0.75 |
| | Prior | 0.62 | 0.06 | 0.04 | 0 | 0.10 | 0 | 0.02 | 0.83 | |
| **Palestine[c]** | Posterior | 0 | 0.01 | 0.02 | 0 | 0.16 | 0 | 0 | 0.19 (0.14-0.19) | 0.89 |
| | Prior | 0 | 0.01 | 0.01 | 0 | 0.08 | 0 | 0 | 0.09 | |
| **Qatar** | Posterior | 0.07 | 0.01 | 0.01 | 0 | 0.23 | 0 | 0.04 | 0.37 (0.31-0.38) | 0.78 |
| | Prior | 0.05 | 0.01 | 0.01 | 0 | 0.19 | 0 | 0.03 | 0.28 | |
| **Saudi Arabia** | Posterior | 0.09 | 1.1 | 0.31 | 0 | 2.5 | 0 | 0.34 | 4.3 (2.4-6.0) | 0.80 |
| | Prior | 0.03 | 0.43 | 0.16 | 0 | 0.88 | 0 | 0.15 | 1.6 | |
| **Syria[c]** | Posterior | 0.06 | 0.03 | 0.41 | 0 | 0.39 | 0 | 0.01 | 0.90 (0.54-1.4) | 0.68 |
| | Prior | 0.05 | 0.01 | 0.16 | 0 | 0.19 | 0 | 0 | 0.42 | |
| **Tunisia** | Posterior | 0.03 | 0.01 | 0.15 | 0 | 0.17 | 0 | 0.04 | 0.40 (0.27-0.50) | 0.61 |
| | Prior | 0.02 | 0.01 | 0.08 | 0 | 0.1 | 0 | 0.02 | 0.23 | |
| **Turkey** | Posterior | 0.03 | 0.08 | 1.4 | 0.16 | 1.2 | 0.02 | 0.13 | 3.0 (2.0-4.1) | 0.93 |
| | Prior | 0.02 | 0.09 | 1.5 | 0.26 | 2.1 | 0.04 | 0.16 | 4.2 | |
| **Turkmenistan** | Posterior | 1.8 | 1.4 | 0.86 | 0 | 0.17 | 0.09 | 0.02 | 4.4 (2.8-5.1) | 0.85 |
| | Prior | 0.84 | 0.58 | 0.34 | 0 | 0.06 | 0.03 | 0.01 | 1.9 | |
| **UAE[c]** | Posterior | 1.4 | 0.07 | 0.05 | 0 | 0.31 | 0 | 0.01 | 1.8 (1.4-2.2) | 0.97 |
| | Prior | 1.4 | 0.06 | 0.04 | 0 | 0.27 | 0 | 0.01 | 1.8 | |
| **Yemen** | Posterior | 0.03 | 0 | 0.19 | 0 | 0.21 | 0 | 0.01 | 0.44 (0.44-0.47) | 0.65 |
| | Prior | 0.02 | 0 | 0.18 | 0 | 0.22 | 0 | 0.01 | 0.44 | |





[a]Prior estimates are from national bottom-up inventories. Posterior estimates are optimized by inversion of TROPOMI observations with uncertainty ranges on total national anthropogenic emissions given in parentheses.  See footnotes in Table 1 for more information on prior and posterior estimates.

[b]Sensitivity of posterior emissions to the TROPOMI satellite observations as determined from the diagonal elements of the reduced averaging kernel matrix (averaging kernel sensitivity). The sensitivity measures the ability of TROPOMI observations to determine the posterior solution independently of the prior estimate, ranging from 0 (not at all) to 1 (fully).

[c]We have limited confidence in separating national emissions between Palestine, Jordan, and Israel; Syria and Iraq; and UAE and Oman in the inversion due to high posterior error correlations. See more details in Sect. 3.2 and Fig. 7.



**Table 3**. Sub-sectoral gas emissions from top emitting countries.[a]

|  | Upstream (Tg a$^{-1}$) | Midstream (Tg a$^{-1}$) | Downstream (Tg a$^{-1}$) |
|---|---|---|---|
| Algeria | 1.6 | 0.29 | 0.20 |
| Turkmenistan | 0.67 | 0.23 | 0.64 |
| Saudi Arabia | 0.02 | 0.46 | 0.38 |
| Iran | 0.18 | 0.15 | 0.68 |

[a]Posterior gas emission estimates from inversion of TROPOMI data, separated by subsector using gridded information from the UNFCCC-based GFEIv2 inventory. Upstream includes exploration, production, and processing. Midstream includes transmission and storage. Downstream includes distribution to consumers. The sum of upstream, midstream, and downstream emissions adds up to the posterior total gas emissions listed in Table 2 for each country.

**Table 4**. Correlation coefficients ($r$) between upstream emissions and activity metrics. [a]

|  | Oil | Gas | Oil + gas |
|---|---|---|---|
| Production rates | -0.16 | 0.14 | -0.23 |
| Well counts | 0.25 | 0.19 | 0.32 |
| Production rates + well counts[b] | 0.26 | 0.28 | 0.28 |

[a]The correlation is calculated using posterior oil/gas upstream emissions and activity data for 12 individual countries listed in Fig. 11.

[b]A multiple linear regression using two explanatory variables (production rates, well counts) to fit the posterior oil/gas emissions.