# Peer review of "Satellite quantification of methane emissions and oil/gas methane intensities from individual countries in the Middle East and North Africa: implications for climate action"

_EGUsphere, 2022_

## Referee Comment (RC1)

Review: Satellite quantification of methane emissions and oil/gas methane intensities from individual countries in the Middle East and North Africa: implications for climate action

ACP Chen et al. January 2023

This is a very interesting and highly policy relevant paper which provides important new insights into the attribution of methane emissions to sector sources in the previously little studied region of the Middle East. I am not an expert on inverse modelling myself but rather on bottom-up modelling and mitigation strategies and will therefore limit my comments to these aspects of the paper. As a bottom-up modeller I find it very encouraging that the quantification of methane emissions using top-down methods (satellites and surface flask measurements) is now narrowing down to the country and sector level. This paper is a good example of this. The paper is well written and easy to follow and I support publication but have a few questions and remarks.

As far as I understand, the authors identify the individual source sectors using prior distributions of emissions from bottom-up inventories complemented with a spatial pattern identifying the type of activities on the ground. To me it is not completely clear how this very fine resolution to individual source sectors was made and how the individual source sectors were identified, e.g., between upstream oil, upstream gas, midstream gas, and downstream gas. I would wish for more clarity on this.

I find the Table 2 very interesting to compare with bottom-up estimates. In this context, I wonder if it would be possible to further split the "Gas" column by upstream and downstream emissions? On p. 10 row 404, I was surprised to see such high emissions from midstream gas transmission from offshore platforms in Saudi Arabia. I would not expect such high emissions from offshore pipelines due to oxidation in the water column, but it refers perhaps to on-shore storage facilities?

I'm intrigued that top-down monitoring now confirms the importance of management practices and technology status for emissions from the oil and gas sector. And also that these are much more important determinants for methane emissions than production quantity or number of wells drilled. It should however be acknowledged that differences in management practices is something that the bottom-up model community has attempted to capture through e.g., simulation of associated gas flows using country-specific information on gas recovery and venting/flaring rates https://iopscience.iop.org/article/10.1088/1748-9326/aa583e. These results are of course much cruder and more uncertain than the results you now put forward in this study.

Changes primarily in management practices, with increases in recovery of associated gas and increased flaring rather than venting of unrecovered gas in several countries are primary reasons why methane emissions from IIASA's GAINS model https://iopscience.iop.org/article/10.1088/2515-7620/ab7457 show a declining trend over time for methane emissions from global oil production:

[Figure]

**Methane emissions from global oil and gas sector in IIASA's GAINS model**

Legend:
- Oil&Gas (July 2022)
- Oil&Gas (ERC, 2020)
- Oil&Gas (ERL, 2017)
- Oil production systems
- Natural gas upstream
- Unconventional gas upstream
- Gas downstream

Hence, not all bottom-up inventories rely fully on activity metrics coupled with constant default emission factors. Some try to find methods to better reflect variations in management practices and technology status as well. The recent advancements in top-down technologies have however brought us a big step forward in better understanding the sources of methane emissions, like the Chen et al. study nicely shows. We now need to feed this knowledge into improved methods for bottom-up inventories, which are likely to remain the primary basis for political negotiations on mitigation targets, at least in the foreseeable future (or until we have the technology to continuously monitor site-level methane emissions globally).

---

## Community Comment (CC1)

This is very interesting work. However, I am confused about the relationship between the "emissions factors" shown in Fig. 10 and the "methane intensities" shown in Fig. 13.

I believe Fig. 10 shows:

$$EF_{oil} = \frac{m_{ch4,oil}}{M_{oil}} \tag{1}$$

$$EF_{NG} = \frac{m_{ch4,NG}}{M_{NG}} \tag{2}$$

where m is mass flux emitted to atmosphere determined from this work's TROPOMI + EDGAR posterior, and M is mass sent to market determined from the bottom-up EIA inventory.

Then is Fig. 13 methane intensity:

$$MI = \frac{m_{ch4,oil} + m_{ch4,NG}}{M_{NG}} = EF_{oil}\left(\frac{M_{oil}}{M_{NG}}\right) + EF_{NG} \tag{3}$$

or does methane intensity have an additional term, such as:

$$MI = \left[EF_{oil}\left(\frac{M_{oil}}{M_{NG}}\right) + EF_{NG}\right] + \frac{m_{flared}}{M_{NG}} \tag{4}$$

where $m_{flared}$ is methane emitted to the atmosphere not as methane but as combustion products, determined from VIIRS satellite using Elvidge (2016) methodology? In other words, does the emissions factor count only the methane that escapes flaring, whereas the methane intensity counts even the methane that gets combusted during flaring?
E.g. why does Iraq have much smaller emissions factors than Turkmenistan in Fig. 10, but much larger methane intensities in Fig. 13? Is it just because Iraq has a larger ratio $(M_{oil}/M_{NG})$ shown in Fig. 11? Or is it additionally because Iraq has a larger $(m_{flared}/M_{NG})$ shown in Fig. 12b?

The present manuscript suggests methane intensity follows Eq. 3 in some sections and Eq. 4 in others. The abstract suggests Eq. 3, because it doesn't mention VIIRS and says "the methane intensity in most countries is considerably higher...[reflecting] incomplete flaring of gas." The figure captions also suggest Eq. 3. Fig. 10 caption says "Emission factors for upstream activity...The emissions factors represent the amount of methane emitted per unit of oil or gas produced, following the definition of IPCC (2006)." Which is very similar langauge to Fig. 13 caption "The methane intensity is defined as the amount of methane emitted from oil/gas upstream activities per unit of methane gas produced."
But the main text of section 3.4 (sentences beginning on lines 536, 548) and the inclusion of Fig. 12b suggests Eq. 4.
I recommend you explicitly define Eq. 3 (or similar) explicitly in the manuscript, and reword the sections that appear to contradict that equation.

It's also difficult to determine these equations from the citations. IPCC (2006) and Elvidge (2016) are cited but missing from the references. Which section of IPCC (2006) defines the emissions factor? The methane intensity is defined in OGCI (2021), but that source is also unclear, as p. 17 of that source seems to count emissions sources fugitive leaks, venting, and flaring, but does not distinguish complete flaring from incomplete flaring.

---

## Author Comment (AC1)

We thank the reviewers for their detailed suggestions and comments on the manuscript. Below, we have replied to each review and have detailed the corresponding edits that we have made to the manuscript. We have listed out the reviewer comments in black *italic* and the replies in blue.

**Referee Comment #1: Dr. Lena Höglund-Isaksson**

*Review: Satellite quantification of methane emissions and oil/gas methane intensities from individual countries in the Middle East and North Africa: implications for climate action ACP Chen et al. January 2023 This is a very interesting and highly policy relevant paper which provides important new insights into the attribution of methane emissions to sector sources in the previously little studied region of the Middle East. I am not an expert on inverse modelling myself but rather on bottomup modelling and mitigation strategies and will therefore limit my comments to these aspects of the paper. As a bottomup modeller I find it very encouraging that the quantification of methane emissions using topdown methods (satellites and surface flask measurements) is now narrowing down to the country and sector level. This paper is a good example of this. The paper is well written and easy to follow and I support publication but have a few questions and remarks.*

We thank the reviewer for the expert insights from bottom-up perspectives. All points have been addressed as below.

**1**.*As far as I understand, the authors identify the individual source sectors using prior distributions of emissions from bottomup inventories complemented with a spatial pattern identifying the type of activities on the ground. To me it is not completely clear how this very fine resolution to individual source sectors was made and how the individual source sectors were identified, e.g., between upstream oil, upstream gas, midstream gas, and downstream gas. I would wish for more clarity on this.*

We described how we aggregate grid emissions to individual sectors and countries, and how individual source sectors are identified in detail on page 8 lines 311-324:

'The posterior GMM state vector ($n \times 1$) can be readily mapped on the $p$ native 0.25°×0.3125° grid cells of the inversion domain using the GMM-generated weighting of each Gaussian on that grid as represented by a matrix $\mathbf{W_1}$ ($p \times n$). The contributions from each of $q$ emission sectors (Table 1) to the emissions in individual grid cells are taken from the prior inventories to produce a matrix $\mathbf{W_2}$ ($pq \times n$). We can then apply a summation matrix $\mathbf{W_3}$ ($r \times pq$) to aggregate emissions over $r$ countries and/or sectors of interest. The resulting matrix $\mathbf{W} = \mathbf{W_3}\mathbf{W_2}$ ($r \times n$) thus represents the linear transformation from the posterior GMM state vector ($n \times 1$) to a reduced state vector ($r \times 1$) of sectoral emissions from individual countries. The reduced state vector ($\boldsymbol{x_{red}}$), posterior error covariance ($\hat{\mathbf{S}}_{\mathbf{red}}$), and averaging kernel matrix ($\mathbf{A}_{\mathbf{red}}$) are computed as

$$\hat{x}_{red} = \mathbf{W}\hat{x} \tag{6}$$
$$\hat{\mathbf{S}}_{\mathbf{red}} = \mathbf{W}\hat{\mathbf{S}}\mathbf{W^T} \tag{7}$$

$$\mathbf{A}_{\mathbf{red}} = \mathbf{W}\mathbf{A}\mathbf{W}^* \tag{8}$$

where $\mathbf{W}^* = (\mathbf{W^T}\mathbf{W})^{-1}\mathbf{W^T}$ is generalized pseudo-inverse of $\mathbf{W}$ (Calisesi et al., 2005).'

The identification between oil/gas subsectors follows the same procedure and we have added one sentence to better communicate this message on page 10 lines 397-402 (underline part added):

'We further analyze gas emission by subsector including upstream or production (leaks, venting, inefficient flaring), midstream (transmission and storage), and downstream (distribution), using gridded sub-sectoral information from GFEIv2. and Table 3 shows results for the top emitting countries, where the sub-sectoral emissions from individual 0.25º×0.3125º grid cells are summed following the procedure of Sect. 2.6.'

**2**.*I find the Table 2 very interesting to compare with bottomup estimates. In this context, I wonder if it would be possible to further split the "Gas" column by upstream and downstream emissions ?*

We have added a Table in the Supplement to show gas emissions by subsector for all 23 countries in the Middle East and North Africa region.

**Table S4**. National gas emissions by subsector in 2019.

| Country | | Gas (Tg a$^{-1}$) | | |
|---|---|---|---|---|
| | | Upstream | Midstream | Downstream |
| **Algeria** | Posterior | 1.6 | 0.29 | 0.20 |
| | Prior | 0.77 | 0.17 | 0.14 |
| **Bahrain** | Posterior | 0.01 | <0.01 | 0.12 |
| | Prior | 0.01 | <0.01 | 0.04 |
| **Egypt** | Posterior | 0.02 | 0.02 | 0.11 |
| | Prior | 0.01 | 0.01 | 0.05 |
| **Iran** | Posterior | 0.18 | 0.15 | 0.68 |
| | Prior | 0.12 | 0.08 | 0.32 |
| **Iraq** | Posterior | <0.01 | 0.01 | 0.03 |
| | Prior | <0.01 | <0.01 | 0.02 |
| **Israel** | Posterior | 0.01 | <0.01 | 0.02 |
| | Prior | <0.01 | <0.01 | 0.01 |
| **Jordan** | Posterior | <0.01 | 0.01 | 0.11 |
| | Prior | <0.01 | <0.01 | 0.05 |
| **Kuwait** | Posterior | <0.01 | <0.01 | <0.01 |
| | Prior | <0.01 | <0.01 | <0.01 |
| **Lebanon** | Posterior | <0.01 | 0.01 | 0.01 |
| | Prior | <0.01 | <0.01 | <0.01 |
| **Libya** | Posterior | 0.02 | <0.01 | 0.01 |
| | Prior | 0.01 | <0.01 | 0.01 |
| **Mauritania** | Posterior | <0.01 | <0.01 | 0.01 |
| | Prior | <0.01 | <0.01 | 0.01 |
| **Morocco** | Posterior | <0.01 | 0.01 | 0.03 |
| | Prior | <0.01 | 0.01 | 0.03 |
| **Niger** | Posterior | <0.01 | <0.01 | <0.01 |
| | Prior | <0.01 | <0.01 | <0.01 |
| **Oman** | Posterior | 0.04 | 0.02 | 0.04 |
| | Prior | 0.02 | 0.01 | 0.03 |
| **Palestine** | Posterior | <0.01 | <0.01 | 0.01 |
| | Prior | <0.01 | <0.01 | <0.01 |
| **Qatar** | Posterior | <0.01 | <0.01 | 0.01 |
| | Prior | <0.01 | <0.01 | <0.01 |
| **Saudi Arabia** | Posterior | 0.02 | 0.46 | 0.38 |
| | Prior | 0.01 | 0.17 | 0.18 |
| **Syria** | Posterior | <0.01 | <0.01 | 0.01 |
| | Prior | <0.01 | <0.01 | <0.01 |
| **Tunisia** | Posterior | <0.01 | <0.01 | 0.01 |
| | Prior | <0.01 | <0.01 | <0.01 |
| **Turkey** | Posterior | <0.01 | 0.04 | 0.04 |

| | | | | |
|---|---|---|---|---|
| | Prior | <0.01 | 0.04 | 0.05 |
| **Turkmenistan** | Posterior | 0.67 | 0.23 | 0.64 |
| | Prior | 0.26 | 0.1 | 0.24 |
| **UAE** | Posterior | 0.01 | 0.01 | 0.05 |
| | Prior | 0.01 | 0.01 | 0.04 |
| **Yemen** | Posterior | <0.01 | <0.01 | <0.01 |
| | Prior | <0.01 | <0.01 | <0.01 |

**3**. *On p. 10 row 404, I was surprised to see such high emissions from midstream gas transmission from offshore platforms in Saudi Arabia. I would not expect such high emissions from offshore pipelines due to oxidation in the water column, but it refers perhaps to onshore storage facilities ?*

Thanks for making this point. Midstream emissions include transmission and storage (described in the footnotes of Table 3), and we now also explicitly state it on page 10 lines 397-402 (underline part added):

'We further analyze gas emission by subsector including upstream or production (leaks, venting, inefficient flaring), midstream (transmission and storage), and downstream (distribution), using gridded sub-sectoral information from GFEIv2. and Table 3 shows results for the top emitting countries, where the sub-sectoral emissions from individual 0.25º×0.3125º grid cells are summed following the procedure of Sect. 2.6.'

And on page 10 lines 406-408:

'Saudi Arabia relies largely on its offshore production for domestic gas use (EIA, 2020). Transmission from offshore platforms to population centers, including onshore storage (Omara et al., 2023) likely explain the large contribution from midstream emissions (53%).'

**4.***I'm intrigued that topdown monitoring now confirms the importance of management practices and technology status for emissions from the oil and gas sector. And also that these are much more important determinants for methane emissions than production quantity or number of wells drilled. It should however be acknowledged that differences in management practices is something that the bottomup model community has attempted to capture through e.g., simulation of associated gas flows using countryspecific information on gas recovery and venting/flaring rates https://iopscience.iop.org/article/10.1088/17489326/aa583e. These results are of course much cruder and more uncertain than the results you now put forward in this study.*

*Changes primarily in management practices, with increases in recovery of associated gas and increased flaring rather than venting of unrecovered gas in several countries are primary reasons why methane emissions from IIASA's GAINS model https://iopscience.iop.org/article/10.1088/25157620/ab7457 show a declining trend over time for methane emissions from global oil production.*
*Hence, not all bottomup inventories rely fully on activity metrics coupled with constant default emission factors. Some try to find methods to better reflect variations in management practices and technology status as well. The recent advancements in topdown technologies have however brought us a big step forward in better understanding the sources of methane emissions, like the Chen et al. study nicely shows. We now need to feed this knowledge into improved methods for botto*

*mup inventories, which are likely to remain the primary basis for political negotiations on mitiga tion targets, at least in the foreseeable future (or until we have the technology to continuously mo nitor site-level methane emissions globally).*

Thank you for raising this important point! It is encouraging that there are consensual efforts from both top-down and bottom-up studies to reflect impact of management practices. We have added sentences to underscore the recent bottom-up studies on page 13 lines 532-538 (underlined part revised or added):

'Standard bottom-up inventories that solely rely on activity metrics are thus unable to accurately quantify oil/gas emissions. Recent bottom-up studies (Höglund-Isaksson et al., 2017, 2020) have advanced the estimation of methane emissions by additively considering the impact of management practices. Höglund-Isaksson et al (2017) in particular simulated global oil/gas emissions for 1980-2012 with the inclusion of country-specific parameters on associated gas flows reflecting variations in managerial decisions, and arrived at closer consistency with top-down estimates.'

**Referee Comment #2: Dr. Amy Townsend-Small**

*This is a very interesting and useful paper! I learned a lot by reading it. I have a few questions/comments.*

We thank the reviewer for the insightful feedback. We have addressed all points as below.

**1**.*Line 148/Figures 2 and 3 - Here you say Iraq, Libya and Oman have not reported their emissions to UNFCCC since 2000 but what about Algeria and Iran? I know their UNFCCC status and Paris Agreement participation is tenuous. I guess in general it would be interesting to know more about the inventories these countries report! As you refer to, this has implications for the Global Stocktake. I realize this is a minimal aspect of your paper but some of these countries are struggling to make an accurate inventory, and your paper could help here.*

Thanks for the suggestion. We have added a Table in the Supplement on the latest report year of an individual country in the Middle East and North Africa.

**Table S1**. Latest year of individual countries reporting to the UNFCCC.

| Country | Algeria | Bahrain | Egypt | Iran | Iraq | Israel | Jordan | Kuwait | Lebanon | Libya[a] | Mauritania | Morocco |
|---|---|---|---|---|---|---|---|---|---|---|---|---|
| Last inventory year | 2000 | 2000 | 2005 | 2000 | 1997 | 2019 | 2016 | 2016 | 2013 | n/a | 2000 | 2012 |
| Country | Niger | Oman | Palestine | Qatar | Saudi Arabia | Syria | Tunisia | Turkey | Turkmenistan | UAE | Yemen | |
| Last inventory year | 2008 | 1994 | 2011 | 2007 | 2012 | 2005 | 2000 | 2020 | 2010 | 2014 | 2012 | |

[a]To date, the government of Libya has not submitted its national inventory to the UNFCCC.

**2**.*Line 384: Here my previous point comes up again. Do Iran and Libya report emissions to UNFCCC? What about Iraq, where does that bottom-up inventory come from? We know Iraq has some of the highest levels of venting and flaring from satellite observations, but I'm not sure how well they are accounting for these emissions.*

Please see response to comment #1. We also described in detail on the bottom-up inventories used as prior information in the inversion on page 4 lines 143-152:

'Fig. 3 shows the distribution of prior emissions by sector over the inversion domain, Table 1 lists the domain-wide totals, and Table 2 lists totals for individual countries. Oil, gas, and coal emissions are from the Global Fuel Exploitation Inventory (GFEIv2), which uses detailed infrastructure data to spatially allocate on a 0.1°×0.1° grid the national inventories from individual countries reported to the UNFCCC including offshore emissions (Scarpelli et al., 2022). Iraq, Libya, and Oman have not reported their emissions to the UNFCCC since 2000 (Table S1), and for those countries GFEIv2 uses recommended emission factors from the IPCC (2006) Tier 1 method and EIA production statistics (EIA, 2020) to infer national emissions. For other anthropogenic sectors (livestock, landfills, wastewater treatment, rice, and other minor sources), prior emissions are from the EDGARv6 inventory for 2018 (Crippa et al., 2021).'

**3**.*Line 437: Interesting that Iran may have higher than expected emissions from livestock and waste. I wonder what their cattle/other ruminant head count is?*

We have added a Table in the Supplement that lists the head counts of ruminant livestock for the six top-emitting countries in the Middle East and North Africa (including Iran).

**Table S3**. Head counts of primary ruminant livestock for top-emitting countries in 2019. [a]

|  | **Iran** | **Turkmenistan** | **Saudi Arabia** | **Algeria** | **Egypt** | **Turkey** |
|---|---|---|---|---|---|---|
| **Cattle** | 5,241,304 | 2,403,120 | 567,040 | 1,786,351 | 2,809,000 | 17,688,139 |
| **Buffalo** | 147,802 | n/a[b] | | | 142,700 | 184,192 |
| **Sheep** | 41,303,611 | 14,053,574 | 9,419,686 | 29,378,561 | 20,82,000 | 37,276,050 |
| **Goat** | 15,034,487 | 2,375,410 | 3,711,155 | 4,929,069 | 977,000 | 11,205,429 |

[a]Data is from the FAOSTAT (*https://www.fao.org/faostat/en/#data/QCL*).

[b]Not available in the FAOSTAT.

**4**.*Line 476: I know Turkey also has a lot of reservoirs? https://en.wikipedia.org/wiki/Southeastern_Anatolia_Project.*

Thanks for raising this point. We have added sentences to describe reservoir emissions in Turkey on page 12 lines 482-486:

'Turkey has many hydroelectric reservoirs (Lehner et al., 2011) that are a source of methane generally not included in national inventories (Li and Zhang, 2014). A global bottom-up inventory of methane emissions from individual hydroelectric reservoirs (Delwiche et al., 2022), including reservoir surfaces and flow through turbines, found emissions of only 0.03 Tg a$^{-1}$ for Turkey, which is small compared to our national emission estimate of 3.0 (2.0-4.1) Tg a$^{-1}$.'

**5**.*Line 498: A study (by one of you) also showed that older marginal wells are a major contributor to methane emissions: https://www.nature.com/articles/s41467-022-29709-3. I have wondered how many marginal wells there are in some of these countries?*

Enverus-based oil and gas infrastructure mapping database (OGIM; https://essd.copernicus.org/preprints/essd-2022-452/) includes facility-level production for North America, Brazil, Argentina, Norway, and Australia, but such information is very limited or unavailable for countries from (e.g.,) the Middle East and North Africa. We have revised and added text to clarify this point on page 12 lines 506-510 (underline part revised or added):

'We examined the correlation with well counts for the Middle East and North Africa by using the Enverus-based Oil and Gas Infrastructure Mapping (OGIM v1; Omara et al., 2023) database, recognizing that the data are incomplete particularly for new wells which could be the largest emitters (Allen et al., 2022) and also possibly for marginal wells (Omara et al., 2022).'

**6**.*Line 555: Doesn't Qatar do a lot of flaring in the North Field? (Zhan Zhang et al 2021 Environ. Res. Lett. 16 124039). This reduces methane emissions but it creates another problem.*

We agree with the reviewer that high-efficiency flaring in Qatar could help reduce methane emissions in Qatar. We have added a sentence to describe it on page 14 lines 568-570.

'Qatar also supports infrastructure upgrades that improve gas flare efficiency from offshore production and boil-off gas recovery in the LNG chains (QatarGas, 2022).'

**7**.*Line 643: Can the bottom-up inventory be included in the supplement?*

Please see our response to comment #2, and we have further added data portal to these inventories in *Data availability*:

'Oil, gas, and coal emissions from the GFEIv2 inventory are available at *https://dataverse.harvard.edu/dataset.xhtml?persistentId=doi:10.7910/DVN/HH4EUM*. Methane emissions by sector from EDGARv6 are available at *https://edgar.jrc.ec.europa.eu/dataset_ghg60*. Wetland emissions from WetCHARTs v1.3.1 are available at *https://doi.org/10.3334/ORNLDAAC/1915*.'

**Community Comment #1: Dr. Nathan Malarich**

*This is very interesting work. However, I am confused about the relationship between the "emissions factors" shown in Fig. 10 and the "methane intensities" shown in Fig. 13.*

*I believe Fig. 10 shows:*

$$EF_{oil} = m_{oil}/M_{oil} \qquad\qquad (1)$$

$$EF_{NG} = m_{NG}/M_{NG} \qquad\qquad (2)$$

*where m is mass flux emitted to atmosphere determined from this work's TROPOMI + EDGAR posterior, and M is mass sent to market determined from the bottom-up EIA inventory.*

Then is Fig. 13 methane intensity:

$$MI = \frac{m_{oil}+m_{NG}}{M_{NG}} = EF_{oil}\left(\frac{M_{oil}}{M_{ng}}\right) + EF_{NG} \qquad\qquad (3)$$

or does methane intensity have an additional term, such as:

$$MI = \frac{m_{oil}+m_{NG}}{M_{NG}} = EF_{oil}\left(\frac{M_{oil}}{M_{ng}}\right) + EF_{NG} + m_{flare}/M_{ng} \quad (4)$$

*Where $m_{flare}$ is methane emitted to the atmosphere not as methane but as combustion products, determined from VIIRS satellite using Elvidge (2016) methodology? In other words, does the emissions factor count only the methane that escapes faring, whereas the methane intensity counts even the methane that gets combusted during faring?*

*The present manuscript suggests methane intensity follows Eq. 3 in some sections and Eq. 4 in others. The abstract suggests Eq. 3, because it doesn't mention VIIRS and says the methane intensity in most countries is considerably higher... incomplete flaring of gas." The figure captions also suggest Eq. 3. Fig. 10 caption says 'Emission factors for upstream activity...The emissions factors represent the amount of methane emitted per unit of oil or gas produced, following the de_nition of IPCC (2006)." Which is very similar langauge to Fig. 13 caption 'The methane intensity is de_ned as the amount of methane emitted from oil/gas upstream activities per unit of methane gas produced." But the main text of section 3.4 (sentences beginning on lines 536, 548) and the inclusion of Fig. 12b suggests Eq. 4. I recommend you explicitly define Eq. 3 (or similar) explicitly in the manuscript, and reword the sections that appear to contradict that equation.*

Thank you for your interest in our work and the detailed comments. We wish to clarify that posterior upstream emissions used to calculate emission factors and methane intensities include leaks, venting, and inefficient flaring, thus referring Eq. 3 as the reviewer kindly wrote above. As for the discussion of flaring activity, we meant to convey that inefficient flaring (but not flaring) is a methane source. We have revised text to avoid ambiguity on page 13 lines 549-550 (underline part added) and throughout the text:

'High methane intensities reflect leaky infrastructure combined with deliberate venting or inefficient flaring of gas.'

And we explicitly define upstream emissions on page 10 lines 397-402:

'We further analyze gas emission by subsector including upstream or production (leaks, venting, inefficient flaring), midstream (transmission and storage), and downstream (distribution), using gridded sub-sectoral information from GFEIv2. and Table 3 shows results for the top emitting countries, where the sub-sectoral emissions from individual 0.25º×0.3125º grid cells are summed following the procedure of Sect. 2.6.'

We also remove 'and eliminate flaring by 2030' on line 548 in the original manuscript to avoid confusion and now it reads (page 14 lines 565-566):

'Saudi Arabia aims to capture most of its associated gas produced (EIA, 2020) as a part of the World Bank's Zero Routine Flaring Initiative'.

**2.** *E.g. why does Iraq have much smaller emissions factors than Turkmenistan in Fig. 10, but much larger methane intensities in Fig. 13? Is it just because Iraq has a larger ratio ($M_{oil}/M_{NG}$) shown in Fig. 11? Or is it additionally because Iraq has a larger ($m_{flare}/M_{NG}$) shown in Fig. 12b?* (This question was asked in comment #1, and we purposely move it down here to facilitate a better communication.).

Thanks for the comment. As described in Eqs. 2-3 of the reviewer's comment, high amount of methane lost from oil activity ($EF_{oil}M_{oil}$) and low gas production to market ($M_{ng}$) contribute to the larger methane intensity in Iraq than in Turkmenistan. This points to poor management practices in Iraq's oil activity. We have added text to explain it on page 13 lines 554-557:

'Moreover, although Iraq has a smaller gas emission factor than Turkmenistan (Fig. 10), its much larger ratio between oil emissions and gas production to market compensates and contributes to a higher methane intensity, pinpointing inadequate operations in Iraq's oil production.'

**3.** It's also difficult to determine these equations from the citations. IPCC (2006) and Elvidge (2016) are cited but missing from the references. Which section of IPCC (2006) denotes the emissions factor? The methane intensity is denoted in OGCI (2021), but that source is also unclear, as p. 17 of that source seems to count emissions sources fugitive leaks, venting, and flaring, but does not distinguish complete flaring from incomplete flaring.

As for methane intensity, please see our response to comment #1. Section 4.2 of IPCC (2006), titled as '*Fugitive Emissions from Oil and Gas Systems*', denotes the oil/gas emission factors. And thank you for catching the missing references, fixed as below!

'Elvidge, C. D., Zhizhin, M., Baugh, K., Hsu, F.-C., and Ghosh, T.: Methods for Global Survey of Natural Gas Flaring from Visible Infrared Imaging Radiometer Suite Data, Energies, 9, 1–15, doi:10.3390/en9010014, 2016.

IPCC: 2006 IPCC guidelines for national greenhouse gas inventories, prepared by the national greenhouse gas inventories program, in: Vol. 2, chap. 4, edited by: Eggleston, H. S., Buendia, L., Miwa, K., Ngara, T., and Tanabe, K., Institute for Global Environmental Strategies (IGES) on behalf of the IPCC, Hayama, Japan, https://www.ipcc-nggip.iges.or.jp/public/2006gl/index.html, 2006.'

**References (newly added):**

Delwiche, K.B., Harrison, J.A., Maasakkers, J.D., Sulprizio, M.P., Worden, J., Jacob, D.J. and Sunderland, E.M.: Estimating drivers and pathways for hydroelectric reservoir methane emissions using a new mechanistic model. Journal of Geophysical Research: Biogeosciences, 127(8), p.e2022JG006908, https://doi.org/10.1029/2022JG006908, 2022.

Elvidge, C. D., Zhizhin, M., Baugh, K., Hsu, F.-C., and Ghosh, T.: Methods for Global Survey of Natural Gas Flaring from Visible Infrared Imaging Radiometer Suite Data, Energies, 9, 1–15, doi:10.3390/en9010014, 2016.

IPCC: 2006 IPCC guidelines for national greenhouse gas inventories, prepared by the national greenhouse gas inventories program, in: Vol. 2, chap. 4, edited by: Eggleston, H. S., Buendia, L., Miwa, K., Ngara, T., and Tanabe, K., Institute for Global Environmental Strategies (IGES) on behalf of the IPCC, Hayama, Japan, https://www.ipcc-nggip.iges.or.jp/public/2006gl/index.html, 2006.

Lehner, B., Reidy Liermann, C., Revenga, C., Vörösmarty, C., Fekete, B., Crouzet, P., Döll, P., Endejan, M., Frenken, K., Magome, J., Nilsson, C., Robertson, J., Rödel, R., Sindorf, N., and Wisser, D.: High- resolution mapping of the world's reservoirs and dams for sustainable river-flow management, Front. Ecol. Environ., 9, 494–502, https://doi.org/10.1890/100125, 2011.

Li, S., & Zhang, Q.: Carbon emission from global hydroelectric reservoirs revisited. Environmental Science and Pollution Research International, 21(23), 13636–13641. https://doi.org/10.1007/s11356-014-3165-4, 2014.

Omara, M., Gautam, R., O'Brien, M., Himmelberger, A., Franco, A., Meisenhelder, K., Hauser, G., Lyon, D., Chulakadaba, A., Miller, C., Franklin, J., Wofsy, S., and Hamburg, S.: Developing a spatially explicit global oil and gas infrastructure database for characterizing methane emission sources at high resolution, Earth Syst. Sci. Data Discuss., https://doi.org/10.5194/essd-2022-452, 2023.